# DriveMamba: Task-Centric Scalable State Space Model for Efficient End-to-End Autonomous Driving

**Haisheng Su**[1], **Wei Wu**[2], **Feixiang Song**[2], **Junjie Zhang**[2], **Zhenjie Yang**[1], **Junchi Yan**[1,†]

[1] Sch. of Computer Science & Sch. of Artificial Intelligence, Shanghai Jiao Tong University
[2] SenseAuto, [†] Corresponding Author
{suhaisheng,yanjunchi}@sjtu.edu.cn

## Abstract

Recent advances towards End-to-End Autonomous Driving (E2E-AD) have been often devoted on integrating modular designs into a unified framework for joint optimization e.g. UniAD, which follow a sequential paradigm (*i.e.,* perception-prediction-planning) based on separable Transformer decoders and rely on dense BEV features to encode scene representations. However, such manual ordering design can inevitably cause information loss and cumulative errors, lacking flexible and diverse relation modeling among different modules and sensors. Meanwhile, insufficient training of image backbone and quadratic-complexity of attention mechanism also hinder the scalability and efficiency of E2E-AD system to handle spatiotemporal input. To this end, we propose **DriveMamba**, a **Task-Centric Scalable** paradigm for efficient E2E-AD, which integrates dynamic task relation modeling, implicit view correspondence learning and long-term temporal fusion into a single-stage **Unified** Mamba decoder. Specifically, both extracted image features and expected task outputs are converted into token-level sparse representations in advance, which are then sorted by their instantiated positions in 3D space. The linear-complexity operator enables efficient long-context sequential token modeling to capture task-related inter-dependencies simultaneously. Additionally, a bidirectional trajectory-guided ***"local-to-global"*** scan method is designed to preserve spatial locality from ego-perspective, thus facilitating the ego-planning. Extensive experiments conducted on nuScenes and Bench2Drive datasets demonstrate the superiority, generalizability and great efficiency of DriveMamba.

## 1 Introduction

Recent years witness significant progress in Autonomous Driving (AD), promoting the research trend of End-to-End learning, which unifies modular designs of various tasks (*i.e.,* perception, prediction and planning) into a differentiable framework. Benefiting from the data-driven joint optimization, E2E-AD methods showcase great efficiency with convincing performance.

Current advances in E2E-AD [12; 17; 37; 39] mainly follow the sequential Transformer paradigm as shown in Fig. 1 (a) and (b), which is under the manual ordering of "perception-prediction-planning" as modular systems. On one hand, such sequential design will inevitably cause information loss and cumulative errors across successive modules. On the other hand, the diversity and flexibility of task relations are also restricted under this connection, hindering the possible task synergies learning. Instead, ParaDrive [43] increases the module flexibility through introducing a multi-task framework with parallel transformer decoders for specific tasks as Fig. 1 (c). However, it still neglects the diverse task relation modeling between modules and lacks a closed-loop evaluation, thus failing to accurately reflect the actual planning performance under realistic driving conditions.

Corresponding author is also affiliated with Shanghai lnnovation Institute. This work was in part supported by Scientific Research Innovation Capability Support Project for Young Faculty (U40) of the Ministry of Education of China (SRICSPYF-ZY2025019).

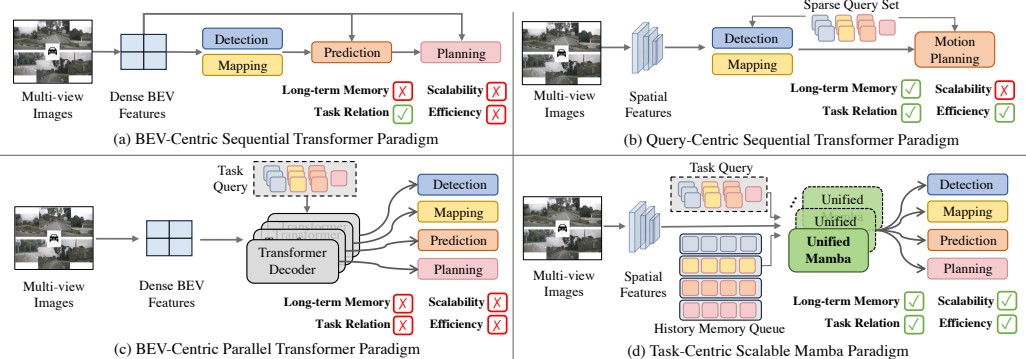

Figure 1: **Comparison of different end-to-end autonomous driving paradigms**. (a) and (b) follow the sequential Transformer paradigm based on dense BEV features [12] and sparse query set [39] respectively. (c) explores the multi-task BEV learning with parallel Transformer decoders [43]. (d) Our proposed **Task-Centric** paradigm learns task-relations dynamically through a unified Mamba decoder, which directly leverages raw sensor inputs and history token memory for long-term spatiotemporal modeling, without construction of expensive BEV features, thus scalable and efficient.

Another challenge of E2E-AD system is handling spatiotemporal input with high efficiency. **BEV-Centric** methods [12; 17; 43] rely on dense BEV feature generation to learn scene representations, which are computationally expensive especially for long-range perception. Besides, BEV-based methods usually perform temporal fusion through storing history BEV features [13; 21], which is impractical for long-term modeling. Meanwhile, the scalability of E2E-AD system is also essential for scaling law exploration with exponential increase of driving data and model parameters. Although recent sparse **Query-Centric** methods [16; 37; 39] are proposed to learn sparse representations, they still have limited scalability owing to the burden of quadratic-complexity attention and lack of efficient interaction order for downstream ego-planning.

We propose **DriveMamba**, a **Task-Centric** scalable paradigm for efficient end-to-end autonomous driving via unified Mamba decoders as shown in Fig. 1 (d). Instead of generating dense BEV features, DriveMamba is built upon sparse representations through image and task tokenization in advance. Then both task queries and image tokens are processed by a unified decoder to capture *dynamic task-task relations* and *implicit task-sensor correspondence* simultaneously in a parallel manner. Meanwhile, a first-in-first-out memory queue is generated to store the history task queries for temporal modeling during the streaming process. Inspired by the adoption of Mamba [8] in NLP domain with its distinct selective state space model (SSM), [26; 51] verify its success in vision tasks which leverage multi-directional SSMs for enhanced 2D image processing. Naturally, in this paper, we investigate its applicability to E2E-AD. Specifically, we introduce a unified Mamba decoder for parallel task relation modeling and view correspondence learning. To facilitate better spatial locality preservation from ego-perspective, a bidirectional **Trajectory-Guided** *"local-to-global"* scan method is designed to handle the spatiotemporal tokens (*i.e.,* Task and Sensor) for long-context modeling with high efficiency. Thanks to its linear-complexity operator, which offers a significant reduction of memory usage than Transformer, our proposed unified Mamba decoder is seamless to scale up through simple stacking. After unified decoding and query enhancement, task queries are send to different task heads for specific purposes. Notably, our DriveMamba-Tiny exhibits great advantages as an efficient end-to-end planner, achieving 53.54 Driving Score on Bench2Drive and 0.13% Collision Rate on nuScenes datasets respectively, with 17.9 FPS running efficiency. **The highlights are as follows**. We leave related works in Appendix A for page limitation.

- We propose **DriveMamba**, a **Task-Centric** paradigm for end-to-end autonomous driving, which integrates view correspondence learning, task relation modeling and long-term temporal fusion into a **Unified** framework with high efficiency and scalability.
- We design a hybrid spatiotemporal scan method to capture task-related inter-dependencies from trajectory-guided **"Local-to-Global"** and **"Spatial-to-Temporal"** bidirectionally, which preserves better spatiotemporal locality for ego-planning.
- Extensive experiments are conducted on nuScenes [2] and Bench2Drive [15] for both open and closed-loop planning evaluation, underscoring its prominent efficiency and scalability.

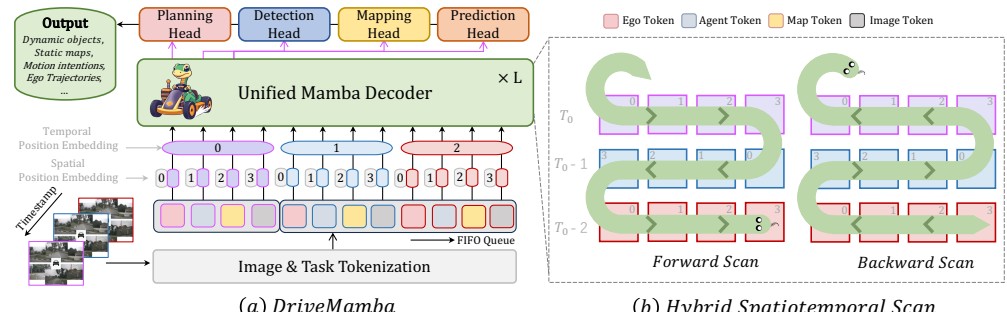

(a) DriveMamba        (b) Hybrid Spatiotemporal Scan

Figure 2: **Framework of DriveMamba**. The multi-view images are encoded into token-level feature sequence and the spatiotemporal queries for different tasks are initialized respectively. Then we adapt the unified Mamba decoder with bidirectional serialization for simultaneous view correspondence learning, task relation modeling and long-term temporal fusion.

## 2 METHOD

### 2.1 PRELIMINARIES

The SSM-based model, Mamba [8], is a discrete variant of continuous system, which establishes a mapping between inputs $x(t) \in \mathbb{R}^M$ and output $y(t) \in \mathbb{R}^M$ through a hidden state vector $h(t) \in \mathbb{R}^N$. This system can be preliminarily expressed as [9]:

$$h'(t) = \mathbf{A}h(t) + \mathbf{B}x(t), \quad y(t) = \mathbf{C}h(t), \tag{1}$$

where $\mathbf{A} \in \mathbb{R}^{N \times N}$ represents the learnable evolution parameters and $\mathbf{B} \in \mathbb{R}^{N \times 1}$, $\mathbf{C} \in \mathbb{R}^{1 \times N}$ are the projection parameters. The zero-order hold (ZOH) method is used to convert $\mathbf{A}$ and $\mathbf{B}$ into discrete parameters $\overline{\mathbf{A}}$ and $\overline{\mathbf{B}}$ with a time scale parameter $\Delta$, which can be defined as follows:

$$\overline{\mathbf{A}} = \exp(\Delta\mathbf{A}), \quad \overline{\mathbf{B}} = (\Delta\mathbf{A})^{-1}(\exp(\Delta\mathbf{A}) - \mathbf{I}) \cdot \Delta\mathbf{B} \tag{2}$$

Compared to previous SSMs, Mamba enhances contextual awareness by introducing a dynamic Selective Scanning Mechanism, abbreviated as S6. Specifically, its parameters $\mathbf{B} \in \mathbb{R}^{B \times M \times N}$, $\mathbf{C} \in \mathbb{R}^{B \times M \times N}$ and $\Delta \in \mathbb{R}^{B \times M \times D}$ are directly obtained from the input $x \in \mathbb{R}^{B \times M \times D}$ for adaptive modulation. Thus, the discretized SSM can be rewritten as:

$$h_t = \overline{\mathbf{A}}h_{t-1} + \overline{\mathbf{B}}x_t, \quad y_t = \mathbf{C}h_t. \tag{3}$$

The original Mamba block is designed for 1D sequence modeling and neglects the spatial awareness when directly adapted to visual tasks. To this aim, Vision Mamba [51] introduces a bidirectional Mamba (B-Mamba) block in Fig. 4(a), which adapts bidirectional sequence modeling for vision-specific applications. This block processes flattened visual sequences through simultaneous forward and backward SSMs, enhancing its capacity for spatially-aware processing. In this work, we extend the B-Mamba block as unified task decoder for end-to-end autonomous driving.

### 2.2 DRIVEMAMBA

Fig. 2 shows the overall framework. We first introduce task-specific Positional Embeddings (PE) to distinguish spatiotemporal tokens from different tasks, sensors and timestamps, which consist of three components: spatial PE, temporal PE and task PE. After that, the token sequence is fed to the proposed DriveMamba with stacked unified decoders for view correspondence learning, task relation modeling and long-term temporal fusion simultaneously following the hybrid scan order.

#### 2.2.1 TOKEN FORMULATION

**Image Tokenization.** To facilitate visual representation learning, multi-view sensor images are firstly encoded by off-the-shelf visual encoders (*i.e.,* ResNet [10], ViT [1], ViM [51]) separately to obtain sensor tokens $\mathbf{T}_{sensor} \in \mathbb{R}^{N_c \times H \times W \times C}$, where $N_c$ is the number of cameras, and $H, W$ indicate the height and width of extracted feature maps after patchification. Hence the sequence length of input sensor tokens is $G = N_c \times H \times W$.

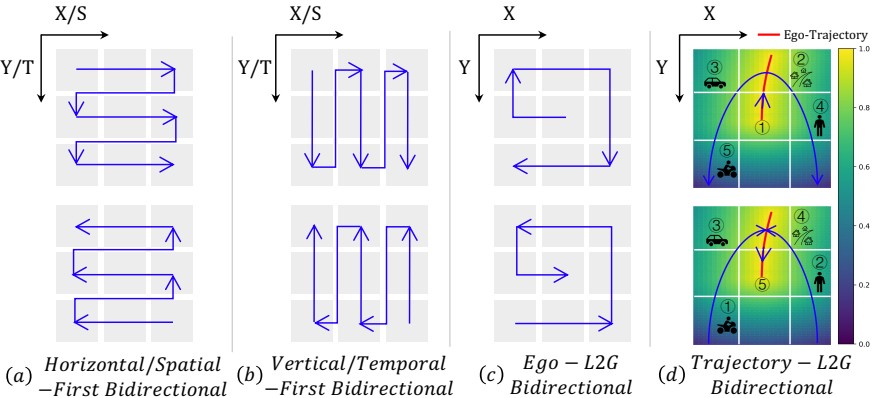

$(a)$ *Horizontal/Spatial −First Bidirectional*   $(b)$ *Vertical/Temporal −First Bidirectional*   $(c)$ *Ego−L2G Bidirectional*   $(d)$ *Trajectory−L2G Bidirectional*

Figure 3: **Different bidirectional scan methods**. Both spatial and temporal scan types are illustrated.

**Task Tokenization.** To conduct interaction modeling between the ego-vehicle and surrounding elements in the driving scene, we pre-define three types of queries to extract task-specific information, namely Agent Queries, Map Queries and Ego Query. Specifically, agent queries represent dynamic objects, which will be used for object detection and motion prediction. Map queries are responsible for online mapping of static elements such as lanes and crossings, etc. Ego query encodes both behavior intention and environmental interactions of the ego-vehicle for planning.

**Token Initialization.** Inspired by DAB-DETR [23], we initialize both sensor tokens and task queries with two parts respectively: semantic embeddings and positional embeddings. The semantic part of agent and map queries are randomly initialized as learnable parameters $\mathbf{Q}_{agent} \in \mathbb{R}^{N_a \times C}$ and $\mathbf{Q}_{map} \in \mathbb{R}^{N_m \times C}$, where $N_a, N_m$ are the number of agent and map queries. And the semantic part of sensor tokens $\mathbf{T}_{sensor}$ are obtained during the image tokenization process, while the ego query's semantic embeddings $\mathbf{Q}_{ego} \in \mathbb{R}^{1 \times C}$ are initialized through encoding the canbus information with an MLP, which is composed of a linear layer and a ReLU activation.

Three types of PE are considered to distinguish spatiotemporal tokens, where Spatial PE encodes the reference position of each token in BEV space, Temporal PE encodes the frame index, and Task PE encodes the task type. The reference positions of agent $\mathbf{P}_{agent} \in \mathbb{R}^{N_a \times 2}$ and map queries $\mathbf{P}_{map} \in \mathbb{R}^{N_m \times 2}$ are initialized uniformly within the pre-defined perception range, while the ego-vehicle position $\mathbf{P}_{ego} \in \mathbb{R}^{1 \times 2}$ is initialized as origin. Then the reference positions $\mathbf{P}$ and timestamps $\mathbf{T}$ are converted into SPE and TPE respectively through sine encoding. Additionally, each token is equipped with one of learnable task embeddings $\mathbf{TE} \in \mathbb{R}^{4 \times C}$. Formally, the **PE** of task token $o$ is:

$$\mathbf{PE}_o = Cat(SE(\mathbf{P}_o), SE(\mathbf{T}_o), \mathbf{TE}_o), \tag{4}$$

where $Cat$ denotes concatenation and $SE$ means sine encoding. Moreover, we adopt an additional depth prediction branch to estimate point-level depth for each sensor token respectively, **instead of relying on the uniformly distributed ray-level 3D positions used in DriveTransformer [16], which cannot be directly employed for token sorting.** Specifically, for each image token $i$ with pixel coordinate $(u_i, v_i)$ of camera $k$, its corresponding location in 3D space is:

$$\mathbf{P}_{sensor}(i, k) = R_k K_k^{-1} [u_i \times d_{i,k}, v_i \times d_{i,k}, d_{i,k}]^T \in \mathbb{R}^3, \tag{5}$$

where $R_k$ and $K_k$ are the extrinsic and intrinsic matrix of camera $k$, and $d_{i,k}$ is the depth value of the predicted depth map. Besides, to model the object movements, the motion-aware layer normalization [42] is adopted for motion compensation implicitly before unified decoding.

### 2.2.2   HYBRID SPATIOTEMPORAL SCAN

To apply the bidirectional Mamba (B-Mamba) block for spatiotemporal input, we extend the original 2D scan into different bidirectional 3D scans as shown in Fig. 3: (a) *Horizontal/Spatial-First*, arranging horizontal/spatial tokens by X-axis/Spatial-dimension BEV location then stacking them vertically/temporally; (b) *Vertical/Temporal-First*, arranging vertical/temporal tokens by Y-axis/Temporal-dimension BEV location then stacking them horizontally/spatially; (c) *Ego-Centric Local2Global*,

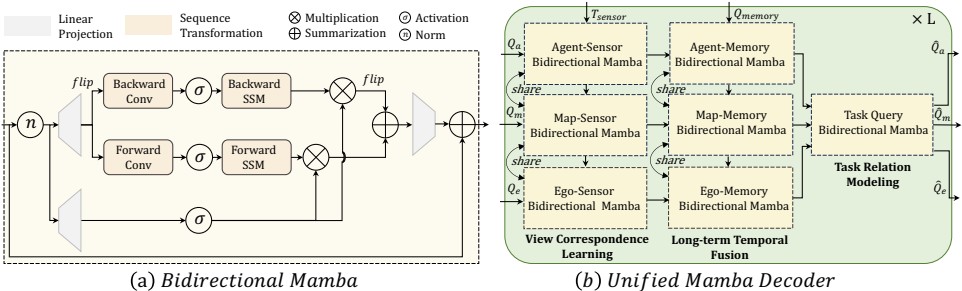

Figure 4: **Detailed structure of Unified Mamba Decoder**. (a) B-Mamba layer is the basic component used for decoding. (b) We illustrate the *divided modeling type* here for clarity.

organizing spatial tokens by BEV location through spiral traversal around the ego origin (refer to Appendix D for more details); (d) *Trajectory-Centric Local2Global*: Considering the diversity and uncertainty of multi-modal ego-planning, we introduce a trajectory-guided attention map to dynamically adjust the scan order based on the relative distances between the interpolated future waypoints $\psi' \in \mathbb{R}^{T'_e \times 2}$ of ego-vehicle and 3D positions $\mathbf{P}_{task}$ of surrounding task queries (*i.e.,* agents and maps) at intermediate layers. The importance $w_i$ of task query $\mathbf{Q}_i$ with reference positions $\mathbf{P}_i$ is:

$$w_i = 1 - \min(\{||\mathbf{P}_i - \psi'_j||_2\}_{j=1}^{T'_e}) / \max(\{\min(||\mathbf{P}_i - \psi'||_2)\}_{i=1}^{N_a+N_m}), \qquad (6)$$

which indicates that **the closest in-path query is prone to obtain the highest importance during the Ego-Env interaction process, conforming to the driving attention distribution**. Moreover, our experiments in Tab. 7 demonstrate that an alternation of *Horizontal/Vertical-First* bidirectional scan across consecutive decoder layers can preserve better spatial locality for view correspondence learning between task queries and sensor tokens, while *Trajectory-Centric Local2Global* bidirectional scan is well designed for task relation modeling and motion planning from ego-perspective. Besides, the *Spatial-First* bidirectional scan is the most effective yet simple for history memory token arrangement. Thanks to the linear complexity of Mamba, our DriveMamba is capable of handling high-resolution and long-term memory tokens efficiently.

**Discussion.** The motivation of our design is to preserve spatial locality suited to different tasks, rather than proposing a more complex scan method. Previous SSM-based models in 2D vision tasks focus on designing well-structured scan patterns for semantic extraction [14; 52]. Furthermore, our trajectory-guided "Local-to-Global" scan considers the interaction order from ego-perspective, which is intuitive for interactive-planning, but neglected in Transformer-based planners (*i.e.*, DriveTransformer).

### 2.2.3 Unified Mamba Decoder

Our unified Mamba decoder (see Fig. 4) consists of three types of B-Mamba layers for different usages: View Correspondence Learning, Task Relation Modeling and Long-term Temporal Fusion. We use *joint modeling type* with a single-layer B-Mamba for each type as discussed in Sec. 3.3.

**View Correspondence Learning.** After bidirectional serialization according to reference positions of task queries $\mathbf{P}_{task}$, we adopt a single B-Mamba layer to allow task queries to directly extract task-specific semantics from raw sensor features with 3D position encoding [25; 34], without construction of dense BEV features and information loss:

$$\hat{\mathbf{Q}}_{task} = \mathbf{VCL}([\mathbf{Q}_{task} \oplus \mathbf{PE}_{task}, \mathbf{T}_{sensor} \oplus \mathbf{PE}_{sensor}]), \qquad (7)$$

where $\oplus$ denotes concatenation, and $\hat{\mathbf{Q}}_{task}$ indicates the updated task query. Cross-view correspondence between desired task outputs and raw sensor inputs can be learned implicitly with linear attention during the joint training.

**Task Relation Modeling.** Benefiting from parallel paradigm of query design, we can further explore dynamic relations among different task queries with an additional B-Mamba layer, thus facilitating the task dependencies modeling (*i.e.,* planning-oriented perception) without manual connections (*i.e.,* sequential paradigm):

$$\hat{\mathbf{Q}}_{task} = \mathbf{TRM}(\mathbf{Q}_{task} \oplus \mathbf{PE}_{task}). \qquad (8)$$

Table 1: **Variants of DriveMamba differ in sizes**, mainly in the visual backbone and decoder layers.

| Model | Decoder | Backbone | Resolution | #Parameters | #Latency |
|-------|---------|----------|------------|-------------|----------|
| Tiny | 3 Layers, 256 Hidden | ResNet-50 | $256\times 704$ | 40.2M | 55.8ms |
| Small | 6 Layers, 256 Hidden | ResNet-50 | $384\times 1056$ | 48.2M | 90.2ms |
| Base | 6 Layers, 512 Hidden | VMamba-B/16 | $384\times 1056$ | 172.2M | 164.3ms |
| Large | 12 Layers, 768 Hidden | ViT-L/16 | $512\times 1408$ | 607.5M | 599.1ms |

Besides, the task relation modeling capability can be seamlessly scaled up with more stacking layers and end-to-end training to capture both intra-class (*i.e.,* Agent-Agent, Map-Map) and inter-class (*i.e.,* Ego-Agent, Ego-Map and Agent-Map) interactions. Meanwhile, the proposed *Trajectory-guided Local-to-Global* scan method also conforms to the natural interaction order of ego-centric planning.

**Long-term Temporal Fusion.** Inspired by [42], we further resort to Query-Centric temporal fusion through maintaining a FIFO-style memory queue to store history queries of different tasks respectively, rather than use history BEV features as [12; 17], which are computationally expensive and storage unfriendly for long-term temporal modeling. The temporal fusion process can be formulated as:

$$\hat{\mathbf{Q}}_{task}^{t_0} = \mathbf{LTF}(\{\mathbf{Q}_{task}^{t}\oplus\mathbf{PE}_{task}^{t}\}_{t=t_0-T_{queue}}^{t_0}), \tag{9}$$

where $t_0$ is the current time-step and $T_{queue}$ is the length of temporal queue. In this way, the temporal fusion can be conducted at each decoder layer and Top-K task queries from decoder output of previous timestamp are pushed into FIFO queues. Moreover, history task queries and positions are transformed into current ego coordinate system and are compensated for object movements with the Motion-aware Layer Normalization (MLN) [42] in advance.

### 2.2.4 ARCHITECTURE DETAILS

As shown in Tab. 1, we design four different variants of DriveMamba architecture with different model sizes. Each variant includes a specific visual backbone and task decoder with fixed number of layers as well as channel dimensions. For the image backbone, both CNN-based (ResNet-50), Transformer-based (ViT-L/16) and Mamba-based (VMamba-B/16) networks are adopted. The 2D image features extracted by the backbone are passed to the unified Mamba decoder together with 3D positional encoding. The unified decoder consists of $L$ layers with $C$ channel dimensions. As for DriveMamba-Tiny, $L$ is set to 3 and $C$ is set to 256. Finally, the decoder outputs both perception, prediction and planning results after parallel decoding with iterative optimization.

### 2.2.5 TASK HEAD

**Object Detection & Motion Prediction.** In DriveMamba, the head for each task generally consists of 2 Fully Connected (FC) layers. After parallel decoding with the unified Mamba decoder, agent tokens pass through a detection head to generate 3D boxes and classes. Subsequently, the agent tokens are augmented with mode embeddings and processed through a motion head, to predict multi-modal trajectories and corresponding probabilities.

**Online Mapping.** Map tokens are initially structured as point-level tokens. With the unified decoder, the updated map tokens are processed by a specific regression head to generate vectorized map instances, where each instance contains 20 points and instance-level class.

**Planning.** DriveMamba employs a series of tokens to represent future waypoints of ego vehicle, and each token encodes a discrete waypoint clue. After task interaction, waypoint tokens are processed by a planning head to predict future trajectory $\psi \in \mathbb{R}^{T_e \times 2}$ with $T_e = 6$ continuous waypoints.

**Iterative Optimization.** Outputs of every block in the unified Mamba decoder are explicitly supervised by specific losses. Each block predicts relative offsets base on the previous outputs, and then updates the task reference points and task position embeddings accordingly for next block.

**Residual Learning.** Before passing the updated task tokens to designed heads for output prediction, we combine the task position embeddings through element-wise addition, so that the task heads can predict positional offsets more accurately.

Table 2: **Open-loop Planning on nuScenes *val* set under ST-P3 metric**. ‡: the usage of ego-status.

| Method | L2 ($m$) ↓ | | | | Collision (%) ↓ | | | | FPS ↑ |
|---|---|---|---|---|---|---|---|---|---|
| | 1$s$ | 2$s$ | 3$s$ | Avg. | 1$s$ | 2$s$ | 3$s$ | Avg. | |
| ST-P3 [11] | 1.33 | 2.11 | 2.90 | 2.11 | 0.23 | 0.62 | 1.27 | 0.71 | 1.6 |
| OccNet [40] | 1.29 | 2.13 | 2.99 | 2.13 | 0.21 | 0.59 | 1.37 | 0.72 | - |
| UniAD [12] | 0.48 | 0.74 | 1.07 | 0.76 | 0.12 | 0.13 | 0.28 | 0.17 | 1.8 |
| VAD-Base [17] | 0.41 | 0.70 | 1.05 | 0.72 | 0.07 | 0.17 | 0.41 | 0.22 | 4.5 |
| BEVPlanner [22] | 0.27 | 0.54 | 0.90 | 0.57 | 0.04 | 0.35 | 1.80 | 0.73 | - |
| DriveMamba-Tiny (**Ours**) | 0.25 | 0.42 | 0.66 | 0.44 | 0.12 | 0.09 | 0.24 | 0.15 | **17.9** |
| DriveMamba-Small (**Ours**) | 0.23 | 0.40 | 0.64 | 0.42 | 0.06 | 0.07 | 0.22 | 0.12 | 11.1 |
| DriveMamba-Base (**Ours**) | 0.22 | 0.40 | 0.63 | 0.41 | 0.05 | 0.06 | **0.21** | 0.11 | 6.1 |
| DriveMamba-Large (**Ours**) | **0.21** | **0.38** | **0.62** | **0.40** | **0.01** | **0.04** | 0.21 | **0.09** | 1.7 |
| AD-MLP [49]‡ | 0.20 | **0.26** | **0.41** | **0.29** | 0.17 | 0.18 | 0.24 | 0.19 | - |
| BEVPlanner++ [22]‡ | **0.16** | 0.32 | 0.57 | 0.35 | **0.00** | 0.29 | 0.73 | 0.34 | - |
| SparseDrive-B [39]‡ | 0.29 | 0.55 | 0.91 | 0.58 | 0.01 | **0.02** | **0.13** | **0.06** | 7.3 |
| ParaDrive [43]‡ | 0.25 | 0.46 | 0.74 | 0.48 | 0.14 | 0.23 | 0.39 | 0.25 | 4.2 |
| DriveTransformer-Small [16]‡ | 0.19 | 0.33 | 0.66 | 0.39 | 0.01 | 0.07 | 0.21 | 0.10 | 10.7 |
| DriveTransformer-Large [16]‡ | **0.16** | 0.30 | 0.55 | 0.33 | 0.01 | 0.06 | 0.15 | 0.07 | 4.5 |
| DriveMamba-Tiny‡ (**Ours**) | 0.19 | 0.35 | 0.59 | 0.38 | 0.07 | 0.09 | 0.24 | 0.13 | **17.9** |
| DriveMamba-Small‡ (**Ours**) | 0.18 | 0.33 | 0.56 | 0.36 | 0.03 | 0.05 | 0.17 | 0.08 | 11.1 |
| DriveMamba-Base‡ (**Ours**) | 0.17 | 0.33 | 0.54 | 0.35 | 0.02 | 0.04 | 0.17 | 0.07 | 6.1 |
| DriveMamba-Large‡ (**Ours**) | **0.16** | 0.30 | 0.52 | 0.33 | **0.00** | 0.03 | 0.16 | **0.06** | 1.7 |

### 2.2.6 END-TO-END TRAINING

**Loss.** Since the tasks in DriveMamba are equally prioritized with no sequential dependencies, a *single-stage* end-to-end training strategy is adopted during the training phase. The training loss is:

$$\mathcal{L} = \sum_{i=1}^{L}(\mathcal{L}_{det}^{i} + \mathcal{L}_{map}^{i} + \mathcal{L}_{depth}^{i} + \mathcal{L}_{motion}^{i} + \mathcal{L}_{plan}^{i}), \qquad (10)$$

where $\mathcal{L}_{det}$ and $\mathcal{L}_{map}$ indicate the perception loss for object detection and online mapping respectively. $\mathcal{L}_{motion}$ corresponds to the motion prediction loss. $\mathcal{L}_{depth}$ is adopted to supervise the depth prediction branch as [34]. $\mathcal{L}_{plan}$ is used for ego-trajectory regression. And $L$ is the block number of the decoder. Meanwhile, vectorized planning constrains identified in [17] such as collision, overstepping and direction are also included in $\mathcal{L}_{plan}$ for regularization. The weight terms $\lambda_{task}$ of different losses are adjusted empirically to ensure the same magnitude.

## 3 EXPERIMENTS

### 3.1 DATASETS AND METRICS

**NuScenes** [2] is a sophisticated outdoor dataset comprising 1000 driving scenarios lasting 20 seconds respectively. It encompasses a total of 1.4M 3D object annotations covering 10 categories, at a frequency of 2Hz. We evaluate the effectiveness of our approach on perception tasks (*i.e.,* object detection and online mapping) using mAP and NDS metrics respectively. And minADE ($m$) and minFDE ($m$) metrics are reported to assess the motion prediction results. Besides, the open-loop planning performance is measured using Collision Rate (%) and L2 Displacement Error (DE) ($m$) metrics, following the conventions in [12; 17].

**Bench2Drive** [15] offers evaluating the multiple capabilities of E2E-AD systems in a closed-loop manner. This benchmark comprises 1,000 clips, covering 44 interactive scenarios, 23 weather conditions, and 12 towns in CARLA v2 [6]. Adhering to the official settings, 950 clips were utilized for training, while the remaining 50 clips were reserved for open-loop evaluation. For closed-loop evaluation, the trained model was deployed in CARLA across 220 test routes, and key metrics e.g. Driving Score (DS), Success Rate (SR), and Efficiency were calculated accordingly.

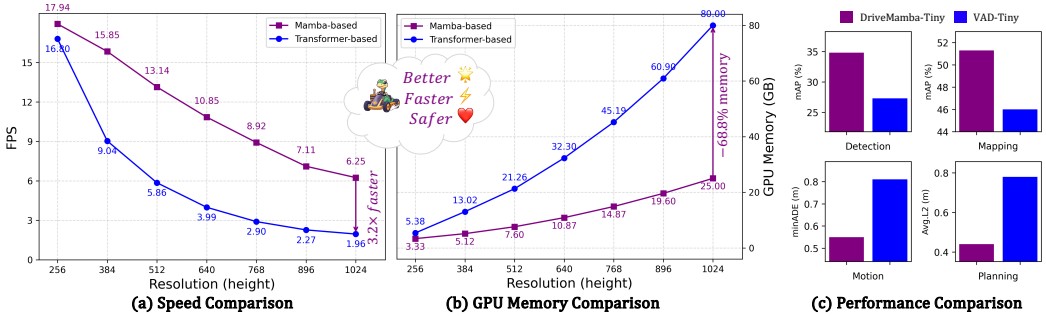

Figure 5: Comparisons between Transformer-based and our Mamba-based E2E-AD methods.

Table 3: **Planning Performance in Bench2Drive [15]**. Avg. L2 is averaged over future 2 seconds.

| Method | Open-loop Metric | Closed-loop Metrics | | | | Latency |
|---|---|---|---|---|---|---|
| | Avg. L2 (m) ↓ | Driving Score ↑ | Success Rate (%) ↑ | Efficiency ↑ | Comfortness ↑ | (ms) |
| AD-MLP [49] | 3.64 | 18.05 | 0.00 | 48.45 | 22.63 | **3.0** |
| UniAD-Base [12] | 0.73 | 45.81 | 16.36 | 129.21 | 43.58 | 663.4 |
| VAD [17] | 0.91 | 42.35 | 15.00 | 157.94 | **46.01** | 278.3 |
| EgoFSD [37] | 0.66 | 60.39 | 31.78 | **180.63** | - | 80.5 |
| DriveTransformer-Large [16] | **0.62** | 63.46 | 35.01 | 100.64 | 20.78 | 211.7 |
| DriveMamba-Tiny (**Ours**) | 0.77 | 53.54 | 27.27 | 137.88 | 20.55 | 55.8 |
| DriveMamba-Small (**Ours**) | 0.75 | 60.78 | 31.82 | 150.46 | 16.52 | 90.2 |
| DriveMamba-Base (**Ours**) | 0.71 | 65.50 | 36.82 | 158.33 | 21.51 | 164.3 |
| DriveMamba-Large (**Ours**) | 0.70 | **66.82** | **37.73** | 152.91 | 18.77 | 599.1 |

Table 4: **Design of Unified Mamba Decoder.** Open-loop planning on nuScenes *val* set is reported.

| Method | Planning L2 (m) ↓ | | | | Planning Coll. (%) ↓ | | | |
|---|---|---|---|---|---|---|---|---|
| | 1s | 2s | 3s | Avg. | 1s | 2s | 3s | Avg. |
| Joint Dec. | 0.25 | 0.42 | 0.66 | **0.44** | 0.12 | 0.09 | 0.24 | **0.15** |
| Divided Dec. | 0.26 | 0.44 | 0.69 | 0.46 | 0.12 | 0.12 | 0.25 | 0.16 |
| w/o VCL | 0.34 | 0.58 | 0.91 | 0.61 | 0.23 | 0.33 | 0.61 | 0.39 |
| w/o LTF | 0.25 | 0.43 | 0.68 | 0.45 | 0.13 | 0.15 | 0.25 | 0.18 |
| w/o TRM | 0.23 | 0.41 | 0.66 | 0.44 | 0.15 | 0.15 | 0.32 | 0.20 |
| w/o IO | 0.25 | 0.44 | 0.71 | 0.47 | 0.42 | 0.31 | 0.41 | 0.38 |
| w/o RL | 0.25 | 0.43 | 0.68 | 0.45 | 0.13 | 0.12 | 0.26 | 0.17 |

Table 5: **Evaluation of Planning-oriented Perception Performance on nuScenes *val* set.** ResNet-50 is adopted as visual backbone.

| Decoder Layers | Detection | | Online Mapping |
|---|---|---|---|
| | mAP ↑ | NDS ↑ | mAP ↑ |
| 3 | 48.6 | 59.9 | 64.7 |
| 6 | 51.2 | 62.5 | 67.0 |
| 9 | 53.6 | 65.1 | 70.4 |
| 12 | 55.1 | 68.3 | 72.2 |

## 3.2 MAIN RESULTS

**Open-loop Planning Evaluation.** As shown in Tab. 2, our DriveMamba exhibits notable advantages over previous works in both planning performance and running efficiency even with shallower visual backbone and fewer decoder layers. Specifically, DriveMamba-Tiny reduces the average L2 error by 42.1% and 38.9% compared with UniAD [12] and VAD [17], along with 11.8% and 31.8% reduction of average collision rate respectively. Besides, benefiting from our efficient sparse representation learning and parallel task decoding, we only consume 55.8 *ms* during inference, achieving 17.9 FPS (10× and 4× faster accordingly). Through deepening the visual encoder and task decoder with more stacking layers, the open-loop planning performance can be further boosted as expected, showcasing the great advantages of our proposed unified yet scalable Mamba paradigm for interactive planning.

**Closed-loop Planning Evaluation.** To further evaluate the interactive planning ability, we also examine the closed-loop planning performance in Bench2Drive [15] as shown in Tab. 3. It can be observed that AD-MLP [49] has a high L2 error and bad closed-loop planning performance using merely ego status as input, which is different from findings in nuScenes [2], demonstrating the behavior diversity in Bench2Drive. Besides, UniAD [12] has a lower L2 error compared to VAD [17] but with worse closed-loop planning performance as discussed in [22]. Recently, DriveTransformer [16] achieves the lowest L2 error with convincing closed-loop driving score. However, our DriveMamba exhibits superior performance with fewer parameters and higher efficiency (*i.e.*, DriveMamba-Base vs. DriveTransformer-Large), underscoring its promising scaling capability.

## 3.3 ABLATION STUDY

**Modular Study.** In Tab. 4, we first study the modular effectiveness of our proposed unified Mamba decoder exhaustively. Three components are ablated respectively, namely View Corresponding

Table 6: **Study of DriveMamba Scalability**. Open-loop perception scalability is evaluated on the nuScenes *val* set, while the closed-loop planning scalability is conducted under the *Dev10* benchmark of Bench2Drive [15] for quick validation. DS: Driving Score, SR: Success Rate.

| Visual Encoder | Decoder Layers | Detection | | Mapping | Planning | |
|---|---|---|---|---|---|---|
| | | mAP ↑ | NDS ↑ | mAP ↑ | DS ↑ | SR ↑ |
| ResNet-50 | 3 | 34.8 | 45.4 | 50.3 | 51.1 | 10 |
| ResNet-101 | 3 | 37.5 | 46.6 | 52.1 | 51.6 | 10 |
| VMamba-B/16 | 3 | 37.1 | 44.8 | 53.5 | 52.6 | 10 |
| ViT-L/16 | 3 | **50.8** | **59.9** | **64.8** | 55.8 | 20 |
| ResNet-50 | 6 | 33.6 | 44.9 | 50.3 | 58.8 | 20 |
| ResNet-50 | 9 | 33.7 | 45.4 | 49.4 | 63.4 | 30 |
| ResNet-50 | 12 | 33.1 | 45.4 | 46.6 | **66.5** | **40** |

Table 7: **Effect of Different Scan Methods.** Both perception and planning performance are reported on nuScenes *val* set with different scan types. EC: Ego-Centric, TC: Trajectory-Centric.

| Spatial | Temporal | Detection | | Mapping | Planning | |
|---|---|---|---|---|---|---|
| | | mAP ↑ | NDS ↑ | mAP ↑ | L2(m) ↓ | Coll. (%) ↓ |
| Horizontal-First | Spatial-First | 31.9 | 41.8 | 48.7 | 0.55 | 0.21 |
| Vertical-First | Spatial-First | 32.8 | 43.4 | 46.3 | 0.52 | 0.22 |
| Horizontal-Vertical | Spatial-First | 33.9 | 44.6 | 50.8 | 0.49 | 0.21 |
| EC Local2Global | Spatial-First | 30.4 | 40.6 | 43.2 | 0.46 | 0.19 |
| TC Local2Global | Spatial-First | 26.0 | 35.1 | 39.8 | 0.45 | 0.17 |
| Hybrid | Spatial-First | **34.8** | **45.4** | **50.3** | **0.44** | **0.15** |
| Hybrid | Temporal-First | 34.7 | 44.3 | 50.2 | 0.48 | 0.16 |

Learning (VCL), Long-term Temporal Fusion (LTF), and Task Relation Modeling (TRM). The VCL layer is the most important operation for randomly initialized queries to extract rich context from sensor tokens. And the LTM conducts efficient temporal modeling with the help of streaming process, while the TRM layer learns dynamic inter-task and intra-task relations for better planning.

**Paradigm Study.** We further study the advantages of our parallel paradigm in many aspects. **(1)** *Modeling Type:* As shown in Fig. 4, three types of B-Mamba layers are illustrated respectively for clarity. However, we compare the performance of joint modeling with a single B-Mamba layer and divided modeling with a shared B-Mamba layer in Tab. 4, where the joint modeling can yield better planning performance without introducing additional parameters. **(2)** *Modeling Order:* Existing methods [12; 17; 39] usually follow the sequential modeling order, which is not conducive to learn diverse task inter-dependencies and thus inferior as in Tab. 2 and 3. **(3)** *Attention Type:* Compared to Transformer-based end-to-end planners, our DriveMamba adopts linear-complexity SSM to learn ego-centric driving attention from sequential task tokens and achieves $3.2\times$ increase in processing speed and requires 68.8% less GPU memory consumption when scaling up input resolution directly as shown in Fig. 5, demonstrating its efficiency and effectiveness in handling long sequential tokens.

**Scalability Study.** In Tab. 6, we study the framework scalability through scaling up both encoder and decoder respectively. **(1)** *Encoder:* When scaling up the visual backbone from ResNet-50, ResNet-101 to VMamba-B/16 and ViT-L/16, we can observe that the open-loop perception performance is increased significantly, while the closed-loop planning performance exhibits a slight improvement. **(2)** *Decoder:* Benefiting from the parallel design, we can scale up the decoder through stacking more layers directly. And we can find that the decoder scaling contributes most on ego-planning while causing a little drop of perception performance, which maybe **because our Task-Centric E2E-AD model learns planning-oriented perception rather than general perception.**

**Planning-oriented Perception Learning.** To evaluate the planning-oriented perception performance, we conduct an additional ablation regarding the perception on Closest In-Path Objects (CIPO) using the nuScenes *val* set, as shown in Tab. 5. CIPO is defined as all objects and lanes within $5m$ around each Ground-Truth future waypoint. The results reveal that CIPO perception performance improves monotonically with additional stacked layers of our unified Mamba decoder, confirming DriveMamba's effectiveness in learning planning-oriented task synergy and dynamic relations.

Table 8: **Initialization of Trajectory Prior.**

| Depth | Detection | | Mapping | Planning | |
|---|---|---|---|---|---|
| | mAP↑ | NDS↑ | mAP↑ | L2(m)↓ | Coll. (%)↓ |
| Origin | 34.8 | 45.4 | 50.3 | 0.44 | 0.15 |
| Uniform | 34.7 | 45.3 | 50.3 | 0.46 | 0.16 |
| Random | 34.6 | 45.1 | 50.2 | 0.45 | 0.14 |

Table 9: **Sensitivity to Prediction Errors.**

| Depth | Detection | | Mapping | Planning | |
|---|---|---|---|---|---|
| | mAP↑ | NDS↑ | mAP↑ | L2(m)↓ | Coll. (%)↓ |
| Incorrect Calibration | 31.7 | 43.1 | 47.5 | 0.46 | 0.16 |
| Noisy Prediction | 32.4 | 43.9 | 48.1 | 0.45 | 0.15 |
| Normal Prediction | 34.8 | 45.4 | 50.3 | 0.44 | 0.15 |
| GT | 42.3 | 50.5 | 54.8 | 0.42 | 0.13 |

Table 10: **Necessity of Trajectory-Centric Scan on Bench2Drive.**

| Scan | Driving Score | Success Rate |
|---|---|---|
| Ego-Centric L2G | 50.27 | 25.45 |
| Trajectory-Centric L2G | 53.54 | 27.27 |

**Design Choice Study.** Moreover, we explore different design choices in Tab. 4 and Tab. 7, including Scan Type, Iterative Optimization (IO) and Residual Learning (RL). To conclude, Horizontal-Vertical spatial scan is useful for task-sensor correspondence learning and Trajectory-Centric Local2Global spatial scan is more reasonable for task relation modeling and ego-centric planning. And **a hybrid of them can lead to ultimate performance**. Meanwhile, Iterative Optimization and Residual Learning also demonstrate their necessities for better performance especially for model scaling up evaluation.

## 3.4 ROBUSTNESS ANALYSIS

**Initialization of Trajectory Prior.** To examine the sensitivity of trajectory prior, we try to ablate different initialization ways of waypoint positions used for token sorting across decoding layers, and the results are shown in Tab. 8, where "Origin", "Uniform" and "Random" indicate zero offset, fixed offset ($\Delta_x$=0, $\Delta_y$=1) and random offset ($\mu = 0$, $\sigma = 1$) initialization of future waypoints of ego vehicle, respectively. And we can find that though with different initialized waypoints, both perception and planning performance exhibit trivial changes, showcasing the training stability of DriveMamba and robustness of our proposed egocentric design for task synergy learning.

**Sensitivity to Depth Prediction Errors.** As shown in Tab. 9, we further evaluate the sensitivity of overall model to various qualities of depth, which is indispensable for sensor token sorting in the view correspondence learning layer during unified decoding. In short, when utilizing the Ground-Truth depth map for sensor token positioning, our DriveMamba can obtain noticeable improvement of perception performance (**+4.9 NDS**) as expected. However, we observe relatively smaller improvement of planning performance (**-0.02% Collision Rate**), demonstrating the robustness of our DriveMamba in handling inaccurate or uncertain detections. Besides, when adding the Gaussian noise to either camera extrinsic parameters (incorrect calibration) or predicted depth map directly, the perception performance drops more significantly than planning performance respectively. It might be because DriveMamba avoids the sequential modeling and construction of BEV features as previous methods, which are sensitive to perception inputs. On the contrary, DriveMamba directly interacts with raw sensor features and thus be able to ignore those failure or noisy inputs and demonstrates better robustness on ego-planning.

**Generalization of Trajectory-Centric Scan.** To evaluate the generalizability of our proposed trajectory-guided scan method, we have also conducted the ablation of Trajectory-Centric L2G scan in Task-Query B-Mamba layer for task relation modeling on Bench2Drive dataset as shown in Tab. 10, which further demonstrates the effectiveness and generalizability of our trajectory-guided scan.

## 4 CONCLUSION

We have proposed a task-centric scalable paradigm based on Mamba architecture for efficient E2E-AD, termed as DriveMamba. DriveMamba integrates dynamic task relation modeling, implicit view correspondence learning and long-term temporal fusion into a unified decoder, which is easy to scale up with simply stacking. Built upon sparse representations, a hybrid spatiotemporal scan method is further designed to capture task-related inter-dependencies without spatial locality loss. Extensive experiments conducted on the nuScenes and Bench2Drive datasets demonstrate the great effectiveness, convincing efficiency and promising scalability of our proposed DriveMamba.

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

## A    RELATED WORK

### A.1    END-TO-END PLANNING

Recent end-to-end planning works can be categorized into implicit and explicit classes. The implicit type of methods [4; 5; 22; 31; 36; 44; 45; 46; 30; 32; 20] perform direct optimization on planning task, which lack interpretability and controllability in realistic applications. [11; 12; 17] enhance the model interpretability through integrating perception, prediction and planning tasks into a unified BEV-based framework explicitly and sequentially. Differently, VAD [17] vectorizes the perception output for interactive planning, thus achieving superior efficiency. ParaDrive [43] removes all explicit task links and proposes a multi-task framework with parallel Transformer decoders. GraphAD [50] utilizes interaction scene graphs to capture object interactions. In addition, some works [3; 31; 33] study how to enhance models' capability through using multiple sensors or incorporating safety-enhanced rules. More recently, there emerges some query-based methods [39; 37; 16] built upon sparse representations. However, most of these methods still overlook the dynamic task-relation modeling and suffer from inefficiency as well as inflexibility to scale up. Particularly, we notice that DriveTransformer [16] is the most similar work which unifies sparse representation, task parallelism and streaming processing into a Transformer decoder for parallel modeling, demonstrating promising model scaling trend. However, it still suffers from limited scalability with quadratic increasing of GPU memory consumption. Besides, DriveTransformer [16] **neglects the importance of interaction order from ego-perspective, which employs global uniform attention with a quadratic-complexity operator, hindering the continuous scalability study and efficient planning-oriented modeling for long-time / high-resolution input and larger models**. Unlike other Transformer-based methods, we did not extensively explore data processing order, but instead integrated the "Task-centric Spatial-Temporal Positional Embeddings", "Trajectory-centric Local-to-Global Scan" and "Query-centric Linear Attention" into a unified Mamba decoder for efficient modeling, which already achieved the best performance with prominent efficiency for further scaling of autonomous driving models.

### A.2    STATE SPACE MODELS

State Space Models (SSMs) have recently demonstrated considerable effectiveness in capturing the dynamic dependencies within language sequences through state space transformation. A structured state-space sequence model is introduced [9] to model long-range dependencies, which inspires various further attempts, such as S5 [35], H3 [7], GSS [29] and Mamba [8]. Differently, Mamba stands out with its selective mechanism using parallel scan (S6). Due to its linear complexity operator, Mamba excels at long sequence modeling compared to Transformer [41] based on quadratic-complexity attention. The great potential of Mamba motivates a series of works [14; 51] in vision tasks, where Vim [51] particularly introduces a bidirectional SSM and position embeddings to learn enhanced 2D image features. Better performance and higher GPU efficiency than Transformer can be obtained on visual downstream tasks like object detection, image classification and semantic segmentation. MambaBEV [47] designs a temporal Mamba model for BEV-level feature fusion and a Mamba-DETR head for 3D object detection. DRAMA [48] adopts Mamba fusion module to fuse camera and LiDAR features in the BEV space, and introduces a Mamba-Transformer decoder to plan a deterministic trajectory. As prior works which employs SSM for efficient sequence modeling in 2D vision tasks (e.g., VisionMamba vs. ViT), we are the first to explore a pure SSM-based decoder for visual End-to-End Autonomous Driving (E2E-AD). Our decoder jointly learns implicit view correspondence, dynamic task relations, and long-term temporal dependencies within a unified, fully sparse, ego-centric design. **To adapt SSM-based framework to E2E-AD, we introduce three key innovations to achieve convincing performance**: (1) Accurate depth estimation of sensor tokens to facilitate spatial positional embeddings construction and reliable spatial/temporal token sorting; (2) Hybrid scan strategies designed for different tasks of various purposes: better spatial locality preservation for semantic perception and trajectory-guided locality preservation for ego-centric planning; (3) Planning-oriented task synergy learning. Extensive ablations and experiments showcase the great efficiency, superiority and scalability of DriveMamba for end-to-end autonomous driving.

## B    EVALUATION METRICS

**Perception.** The evaluation for object detection and online mapping follows standard evaluation protocols [2; 38; 19; 18]. For detection, we use mean Average Precision (mAP), mean Average Error of Translation (mATE), Scale (mASE), Orientation (mAOE), Velocity (mAVE), Attribute (mAAE)

and nuScenes Detection Score (NDS) to evaluate the model performance. For online mapping, we calculate the Average Precision (AP) of three map classes: lane divider, pedestrian crossing and road boundary, then average across all classes to get mean Average Precision (mAP).

**Motion Prediction.** Following the standard motion prediction protocols, we adopt minADE (minimum Average Displacement Error), minFDE (minimum Final Displacement Error) and MR (Miss Rate) as evaluation metrics. Similar to the prior works, these metrics are only calculated within matched TPs, and we set the matching threshold to $1.0m$ in all of our experiments. As for the MR, we set the miss FDE threshold to $2.0m$. Similarly to the prior works [12; 39], we merge the car, truck, construction vehicle, bus, trailer, motorcycle, and bicycle as the vehicle category, and all the motion prediction metrics are measured on the vehicle category only.

**Planning.** We adopt commonly used L2 error and collision rate to evaluate the planning performance. The evaluation of L2 error is aligned with VAD [17]. For collision rate, there are two drawbacks in previous [12; 17] implementation, resulting in inaccurate evaluation in planning performance. On one hand, previous benchmark convert obstacle bounding boxes into occupancy map with a grid size of 0.5m, resulting in false collisions in certain cases, e.g. ego vehicle approaches obstacles that smaller than a single occupancy map pixel [49]. (2) The heading of ego vehicle is not considered and assumed to remain unchanged [22]. To accurately evaluate the planning performance, we account for the changes in ego heading by estimating the yaw angle through trajectory points, and assess the presence of a collision by examining the overlap between the bounding boxes of ego vehicle and obstacles. We reproduce the planning results on our benchmark with official checkpoints [12; 17] for a fair comparison.

## C  IMPLEMENTATION DETAILS

DriveMamba is implemented upon Bench2DriveZoo [15]. Following the previous methods [17], we set the perception range to $60m \times 30m$ longitudinally and laterally, and adopt ResNet-50 as the default backbone to encode image features. DriveMamba plans a 3 seconds (2Hz) future trajectory of ego-vehicle using 2 seconds history information as input. We have four variants of DriveMamba with different sizes of backbone, image resolution and decoder layer as shown in Tab. 1, namely DriveMamba-Tiny, DriveMamba-Small, DriveMamba-Base and DriveMamba-Large. The default number of agent query and map query are set to 900 and $125 \times 20$, while the memory queue length is set to $T_{queue} = 4$ (2 seconds) with Top-K = 256 query propagating continuously as [42]. We train DriveMamba for 90 epochs on nuScenes and 30 epochs on Bench2Drive respectively. All experiments are conducted using 8 Tesla A800 GPUs, utilizing AdamW [28] optimizer and Cosine Annealing [27] scheduler to train DriveMamba with weight decay 0.01 and initial learning rate $2 \times 10^{-4}$. $T'_e$ is set to 30 for trajectory interpolation.

## D  MORE DETAILS

**Formulation of Ego-Centric Local2Global Scan**. For $N \times N$ matrix, the layer $l$ is the minimum distance from the border to the location $(x, y)$:

$$l = min(x, y, N - 1 - x, N - 1 - y), \tag{11}$$

and the index $k$ of location $(x, y)$ can be calculated as:

$$k = \begin{cases} 4l(N-l) + (y-l), & x = l \\ 4l(N-l) + (N-2l-1) + (x-l), & y = N-1-l \\ 4l(N-l) + 2(N-2l-1) + (N-1-l-y), & x = N-1-l \\ 4l(N-l) + 3(N-2l-1) + (N-1-l-x), & y = l \end{cases} \tag{12}$$

**Data Augmentation.** During training phase, we conduct a global data augmentation strategy to improve the model stability. Specifically, random rotation, translation and flipping (Y-axis) are applied to both bounding boxes of agent / map instances and future trajectories of agents / ego-vehicle respectively. Meanwhile, we adjust the camera extrinsic parameters accordingly to ensure spatial alignment between the surround-view images and the noise-perturbed targets.

Table 11: **Comparisons of different training strategies.** E2E training showcases better results.

| Method | Planning L2 ($m$) ↓ | | | | Planning Coll. (%) ↓ | | | |
|---|---|---|---|---|---|---|---|---|
| | 1s | 2s | 3s | Avg. | 1s | 2s | 3s | Avg. |
| End-to-End Training | 0.25 | 0.42 | 0.66 | **0.44** | 0.12 | 0.09 | 0.24 | **0.15** |
| Two-Stage Training | 0.24 | 0.41 | 0.66 | 0.44 | 0.11 | 0.14 | 0.46 | 0.23 |
| Planning-Only Training | 0.25 | 0.44 | 0.69 | 0.46 | 0.16 | 0.18 | 0.25 | 0.20 |
| w/o Dense Supervision | 0.25 | 0.44 | 0.71 | 0.47 | 0.42 | 0.31 | 0.41 | 0.38 |

Table 12: **Study of BEV Size.** Different sizes used for position sorting in 3D space.

| BEV Size | Planning L2 ($m$) ↓ | | | | Planning Coll. (%) ↓ | | | |
|---|---|---|---|---|---|---|---|---|
| | 1s | 2s | 3s | Avg. | 1s | 2s | 3s | Avg. |
| 25×25 | 0.23 | 0.41 | 0.66 | 0.43 | 0.07 | 0.10 | 0.27 | 0.15 |
| 50×50 | 0.23 | 0.41 | 0.65 | **0.43** | 0.02 | 0.05 | 0.24 | **0.10** |
| 75×75 | 0.23 | 0.40 | 0.65 | 0.43 | 0.09 | 0.13 | 0.26 | 0.16 |
| 100×100 | 0.23 | 0.41 | 0.67 | 0.44 | 0.13 | 0.15 | 0.28 | 0.19 |

Table 13: **Necessity of Trajectory-Guided Scan.** Both predicted and Ground-Truth future trajectories are adopted respectively for attention map generation during the task relation modeling process.

| Method | Planning L2 ($m$) ↓ | | | | Planning Coll. (%) ↓ | | | |
|---|---|---|---|---|---|---|---|---|
| | 1s | 2s | 3s | Avg. | 1s | 2s | 3s | Avg. |
| Predicted Trajectory | 0.23 | 0.41 | 0.65 | 0.43 | 0.02 | 0.05 | 0.24 | 0.10 |
| GT Trajectory | 0.17 | 0.26 | 0.35 | **0.26** | 0.03 | 0.05 | 0.13 | **0.07** |

**Generation of Trajectory Prior.** As stated in the token initialization section, each pre-defined task query is equipped with both semantic embeddings and positional embeddings. And we initialize the BEV location of ego query as origin (0, 0). Specifically, DriveMamba employs a series of tokens (e.g. $T_e = 6$) to represent future waypoints of ego vehicle, where each token encodes a discrete waypoint position. Then the future waypoints are iteratively refined with predicted offsets across consecutive decoding layers through planning head, facilitating the adaptive token sorting from ego-perspective.

**Hybrid Combination of Spatiotemporal Scan.** Empirically, we find that an alternation of Horizontal-First and Vertical-First bidirectional scan, named as Horizontal-Vertical, across consecutive decoding layers (i.e. $L_0$: H-First, $L_1$: V-First, $L_2$: H-First, ..., etc) can preserve better spatial locality for **view correspondence learning**, leading to accurate perception. Besides, Trajectory-Centric L2G spatial scan exhibits superior planning performance from ego perspective during the **task relation modeling**. While the spatial-first temporal scan excels at **long-term temporal fusion**. Intuitively, we combine these three scan strategies into a unified decoder for different purposes of B-Mamba layers, termed as hybrid spatiotemporal scan. Moreover, the alternation of H/V-First spatial scan contributes to the training convergence and stability of DriveMamba for structural representation learning from sensor tokens, thus providing high-quality task queries for relation modeling and temporal fusion. And Trajectory-Centric L2G scan can further facilitate ego-planning with adapative interaction order. Each ablation about hybrid scan combination has been carried out with three repeated experiments independently, demonstrating the consistent conclusion.

# E   MORE ABLATION STUDY

**Training Strategies.** As show in Tab. 11, we find that the end-to-end training can achieve superior performance than divided training, which is different to findings of previous methods [12; 17] following sequential paradigm. And the dense supervision contributes to the model convergence and benefits model scaling up. Besides, without perception supervision, the planning-only training can also result in considerable performance through extracting information directly from raw sensors.

**Effect of BEV Size.** The BEV size is important for indexing the task queries in 3D space with instantiated reference positions. As shown in Tab. 12, we compare different BEV sizes of DriveMamba-Small from small to large. And we find that when the BEV size is set to 50×50, our DriveMamba can achieve the best planning performance. The smaller size may cause the indistinguishable importance among different queries, and the larger size will inevitably lead to discontinuity of spatial information.

**Necessity of Trajectory-Guided Scan.** As shown in Tab. 13, we further evaluate the necessity of trajectory guidance for dynamic attention map generation used for position sorting. When utilizing the ground-truth future trajectories for upper-limit evaluation, our DriveMamba-Small can obtain the extremely excellent planning performance ($0.26m$ average L2 error with 0.07% average collision rate), demonstrating that the trajectory-guided scan can contribute to the high-quality query sorting, which is essential for accurate trajectory planning, and vice versa.

Table 14: **Performance comparison of different E2E-AD methods on middle tasks**, including both BEV-Centric, Query-Centric and Task-Centric paradigms. †: Reproduced with official checkpoint.

| Method | Detection | | Online Mapping | | | | Motion Prediction | | |
|---|---|---|---|---|---|---|---|---|---|
| | mAP ↑ | NDS ↑ | $AP_{ped}$ ↑ | $AP_{divider}$ ↑ | $AP_{boundary}$ ↑ | mAP ↑ | minADE ↓ | minFDE ↓ | MR ↓ |
| UniAD [12] | 38.0 | 49.8 | - | - | - | - | 0.71 | 1.02 | 0.15 |
| VAD† [17] | 31.3 | 43.6 | 42.5 | 50.5 | 49.8 | 47.6 | 0.74 | 1.10 | 0.13 |
| SparseDrive [39] | 49.6 | 58.8 | 53.2 | 56.3 | 59.1 | 56.2 | 0.60 | 0.96 | 0.13 |
| DriveTransformer [16] | 49.9 | 59.3 | - | - | - | - | 0.61 | 0.95 | 0.13 |
| **DriveMamba** | 50.8 | 59.9 | 60.4 | 65.8 | 68.1 | 64.8 | 0.55 | 0.76 | 0.08 |

Table 15: **Module runtime statistics**. The inference speed is measured for DriveMamba-Tiny on NVIDIA GeForce RTX 3090 GPU as [17].

| Module | Latency (*ms*) | Proportion (%) |
|---|---|---|
| Backbone | 5.1 | 9.1 |
| Depth Prediction | 4.0 | 7.2 |
| Bidirectional Serialization | 1.2 | 2.2 |
| View Correspondence Learning | 4.4 | 7.9 |
| Long-term Temporal Fusion | 11.6 | 20.8 |
| Task Relation Modeling | 3.9 | 7.0 |
| Task Head Inference | 7.8 | 13.9 |
| Temporal Memory Propagation | 17.8 | 31.9 |
| Total | 55.8 | 100.0 |

**Comparison on Middle-Task Performance.** To study the generalizability of our proposed Drive-Mamba, we also compare the performance of middle tasks on nuScenes validation set as shown in Tab. 14. It can be observed that our proposed DriveMamba surpasses all existing end-to-end autonomous driving methods by a notable margin, including both BEV-Centric [12; 17] and Query-Centric [39] paradigms. The probable reason maybe owing to the proposed effective yet efficient parallel paradigm with Task-Centric scalable Mamba decoding. Diverse task synergies and planning-oriented optimization can be simultaneously leveraged for single-stage end-to-end training.

**Efficiency Analysis.** Finally, we analyze the efficiency of DriveMamba with module runtime statistics as shown in Tab. 15. Among all modules and operations, the long-term temporal fusion and temporal memory propagation occupy the most proportion (52.7%), which contribute to the streaming process for parallel decoding. Thanks to the linear-complexity attention operation and parallel decoding, our DriveMamba achieves great efficiency and scalability.

## F  DISCUSSION

**Comparison to Transformer-based E2E Planners.** Traditional attention-based E2E planners mainly adopt sequential Transformer paradigm based on either dense BEV features [12; 17] or sparse query set [39]. However, sequential design relying on manual order can inevitably cause information loss and cumulative errors across modules. Besides, diverse relation modeling among different modules is also neglected for task-oriented optimization. Moreover, lack of a unified decoder hinders the scalability for building up a large-scale model. Recently, ParaDrive [43] explores a multi-task BEV framework with parallel Transformer decoders to study the necessity of different modules and the impact of their connectivity. However, BEV-based view transformation is computational expensive and training insufficient owing to sparse gradients. Although with elaborate query design, the quadratic complexity attention mechanism adopted in DriveTransformer [16] still suffers from dramatic GPU memory consumption when scaling up gradually. In contrast, our DriveMamba processes spatiotemporal tokens with **linear complexity** and **trajectory-guided hybrid scan**, outperforming both BEV-Centric and Query-Centric end-to-end planners on all tasks of nuScenes dataset (see Tab. 2 and 14).

Table 16: Comparisons of different DriveMamba and DriveTransformer [16] variants over sizes under Bench2Drive *Dev10* benchmark.

| Model | Backbone | #Parameters | Latency | Driving Score |
|---|---|---|---|---|
| DriveTransformer-Small | ResNet-50 | 47.41M | 93.8ms | 45.0 |
| DriveTransformer-Base | ResNet-50 | 178.05M | 139.6ms | 60.5 |
| DriveTransformer-Large | ResNet-50 | 646.33M | 221.6ms | 68.2 |
| DriveMamba-Tiny | ResNet-50 | 42.2M | 55.8ms | 51.1 |
| DriveMamba-Small | ResNet-50 | 48.2M | 90.2ms | 58.8 |
| DriveMamba-Base | VMamba-B/16 | 172.2M | 164.3ms | 68.5 |
| DriveMamba-Large | ViT-L/16 | 607.5M | 599.1ms | **70.3** |

Furthermore, DriveMamba achieves a $3.2\times$ increase in processing speed and requires 68.8% less GPU memory for long-time/high-resolution image sequences (see Fig. 5), demonstrating its efficiency and effectiveness in handling high-resolution sensor data and performing long-horizon tasks.

**Comparison to DriveTransformer [16].** Unlike other Transformer-based methods, we did not extensively explore data processing order (*i.e.*, Sequential or Parallel), but instead integrated the "Task-centric SpatialTemporal Positional Embeddings", "Trajectory-centric Local-to-Global Scan" and "Query-centric Linear Attention" into a unified decoder for parallel modeling, which already achieved the best performance with promising efficiency for E2E-AD model scaling. And our propose DriveMamba mainly differs from previous DriveTransformer in design motivation, token initialization, modeling type and attention pattern.

Specifically, we perform **View Correspondence Learning** between 2D sensor tokens and 3D task tokens through point-level depth estimation and hybrid token sorting in 3D space, which are fed to the linear-complexity SSM layer for efficient feature extraction. However, **Sparse Representation** adopted in DriveTransformer is constructed through combining the 2D image features and 3D ray-level positional embeddings following PETR series [24; 25], then the task tokens decode semantics from the pseudo 3D features with transformer layers. Besides, we conduct **Task Relation Modeling** through introducing the trajectory-guided "Local-to-Global" geometric prior, which is demonstrated to facilitate the ego-centric interactive planning with extensive experiments. Differently, **Task Parallelism** proposed in DriveTransformer simply models the task relations through global-wise quadratic-complexity self-attention, neglecting the importance of interaction order from ego-perspective, leading to inferior planning performance and efficiency. Further, we conduct the **Long-term Temporal Fusion** through bidirectional SSM, which has shown great capacity of sequential modeling in natural language processing (NLP) domain. Differently, **Streaming Processing** in DriveTransformer simply follows StreamPETR [42] to conduct computationally expensive self-attention in temporal domain. In sum, we propose a pure, unified yet scalable SSM-based decoder to learn implicit view correspondence, dynamic task-relations and long-term temporal dependencies with the ego-centric design for End-to-End Autonomous Driving (E2E-AD).

Quantitatively, we compare different variants of DriveMamba with DriveTransformer as shown in Tab. 16, which showcases great efficiency of our DriveMamba with superior planning performance. For example, DriveMamba-Tiny outperforms DriveTransformer-Small by 6.07 in Driving Score while reducing latency by 40.5% (55.8ms vs. 93.8ms). And DriveMamba-Small is comparable with DriveTransformer-Base with fewer parameters and higher efficiency.

**Comparison to DRAMA [48].** Recent works such as DRAMA also proposes an end-to-end motion planner with Mamba design, which seeks for a combined Mamba-Transformer decoder to improve the planning performance. Specifically, DRAMA targets at using a Mamba Fusion module to encode multi-modal BEV features from both cameras and LiDAR. Besides, its decoder follows a Mamba-Transformer structure to enhance the attended features for planning output. Differently, our DriveMamba introduces a pure Mamba decoder for visual end-to-end autonomous driving, which integrates the view correspondence learning, task-relation modeling and long-term temporal fusion into a unified decoder with fully sparse query design. State-of-the-art performance on different tasks (*i.e.,* perception, prediction and planning) are achieved on both open-loop and closed-loop benchmarks with great efficiency.

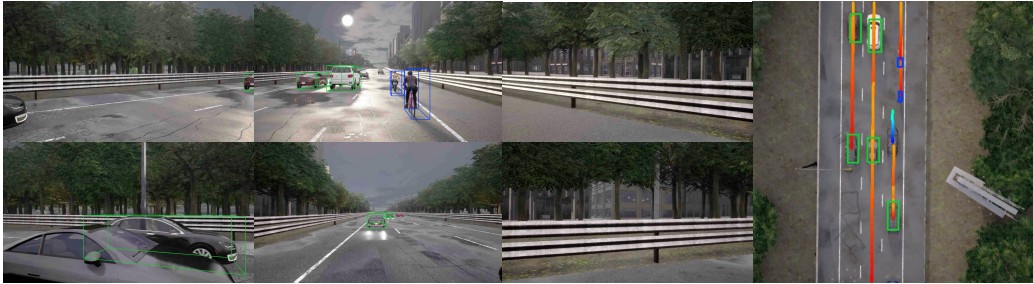

(a) Ego vehicle brakes when encounters slow-moving hazard blocking part of the lane and the side lane is not available

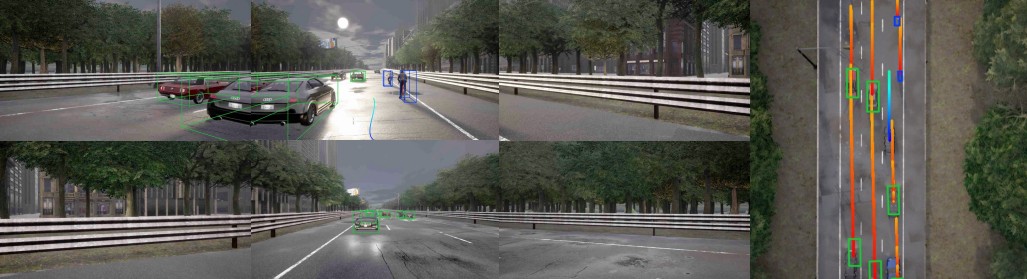

(b) Ego vehicle maneuvers next to a lane of traffic moving in the same direction to avoid it when the side lane is available

Figure 6: Qualitative results of DriveMamba on closed-loop routes of an representative interactive scene - "HazardAtSideLane". Task outputs of DriveMamba are illustrated including perception, motion, and planning trajectories.

## G  LIMITATIONS AND SOCIAL IMPACT

**Limitations.** Though our proposed DriveMamba achieves enormous advancement and showcases great potential of parallel Mamba paradigm for end-to-end autonomous driving, the excessive entanglement of the framework will also constrain flexible problem localization and debugging. A combination of joint and divided modeling with parallel decoders is worth further research.

**Social Impact.** DriveMamba could be easily deployed on mass-produced car chips with different limitations of computing resources, and thus can be served as a plug-and-play software to assist human drivers in decision-making and safe driving.

## H  VISUALIZATION

As show in Fig. 6, we visualize the closed-loop results on the Bench2Drive test routes. An typical interactive scenario named as "HazardAtSideLane" is illustrated across different timestamps. Specifically, in Fig. 6 (a), the ego vehicle decides to brake when encounters slow-moving hazard blocking part of the lane, meanwhile the side lane is not avaliable owing to the congested traffic. And in Fig. 6 (b), when the side lane is available, the ego vehicle start to maneuver next to a lane of traffic moving forward to avoid the hazard for efficient driving, which demonstrates the great interactive planning capacity of our DriveMamba considering both safety, comfort and efficiency.

