# OpenReview forum: "DriveMamba: Task-Centric Scalable State Space Model for Efficient End-to-End Autonomous Driving"
_ICLR.cc/2026/Conference — ICLR 2026 Poster_

### Official Review · Reviewer_qoTq · 2025-10-24

**Soundness:** 2
**Presentation:** 3
**Contribution:** 2
**Rating:** 6
**Confidence:** 5

**Summary:**

This paper proposes DriveMamba, a novel task-centric and scalable paradigm for end-to-end autonomous driving that replaces traditional sequential or Transformer-based modules with a unified Mamba decoder built on selective state-space models for linear-time complexity. Its key contributions include a sparse token representation that integrates image features and task queries (for perception, prediction, and planning) with 3D positional encoding, a hybrid spatiotemporal scan method—featuring a trajectory-guided "local-to-global" scan—that preserves spatial locality and enhances ego-centric planning, and a unified architecture that simultaneously learns view correspondence, dynamic task relations, and long-term temporal fusion. Extensive experiments on nuScenes and Bench2Drive show that DriveMamba achieves state-of-the-art planning accuracy and efficiency.

**Strengths:**

1. Brief framework: The paper presents a scalable and unified framework for end-to-end autonomous driving, characterized by its simplicity, effectiveness, and elegant design.

2. Clear presentation and comprehensive experiments: The paper is clearly written and well-structured, with extensive experiments that convincingly demonstrate the effectiveness of the proposed approach.

**Weaknesses:**

1. **Novelty**: This paper largely follows the framework of DriveTransformer, sharing a similar architecture, experimental setup, and overall structure. The main modification lies in replacing the Transformer module with Mamba, which leads to only marginal improvements in performance and efficiency. Therefore, while the paper does not exhibit major flaws, its contribution is relatively incremental and may receive only a moderate level of interest.

**Questions:**

See weaknesses.

---

> ### Author Response · Authors · 2025-11-21
> **Response to Reviewer qoTq**
>
> Thanks for your positive evaluations and valuable feedback sincerely, we provide the following reponse aiming to address your concerns:
>
> > **Q1: Concern about the novelty of proposed method.**
>
> **Response:** Thanks for your question. We would like to further clarify the main contributions of DriveMamba, which aims to improve the scalability and efficiency of current driving systems for end-to-end planning. Specifically, we are the first to apply the 1D Mamba model to the 3D driving space for unified decoding of scene representation, long-term temporal information and diverse task relation. And we explicitly introduce point-level geometry guidance and design three customized scan methods to faciliate the token sorting for different usages, thus preserving better spatial locality from different perspectives (e.g. objects and ego vehicle). Exhaustive ablations confirm the necessity of such hybrid interaction order for efficient task-oriented learning, which are neglected in Transformer-based methods. Besides, we innovatively define the CIPO (Closest In-Path Objects) perception metrics used for end-to-end autonomous driving, which can intuitively reflect and explain the effectiveness of planning-oriented learning, rather than general perception purpose as previous methods (i.e. UniAD, VAD, ParaDrive, DriveTransformer).
>
> To sum, our DriveMamba proposes a scalable, explainable yet unified decoder with memory-friendly linear attention, which provides valuable insight for future research development on ego-centric task synergy learning. Extensive experiments and comparisons also demonstrate the intuitive and nontrivial improvement of DriveMamba in enhancing the capacity of handling long-term and high-resolution multi-view sensor inputs (**3.2x speed increase and 68.8% less memory usage** at higher resolutions), which is essential for streaming driving scene understanding, reasoning and planning, instead of simply replacing the Transformer decoder with Mamba decoder. We will highlight the motivation and contribution of DriveMamba in the revised version.

---

> > ### Comment · Reviewer_qoTq · 2025-11-27
> >
> > Thank you for the author's reply. Overall, this work is quite solid. I still have some concerns about novelty, but overall, I still have a positive assessment of this paper.

---

> > > ### Author Response · Authors · 2025-11-27
> > >
> > > Thanks for your positive feedback. We appreciate your advice to make the manuscript better!

---

### Official Review · Reviewer_RMQD · 2025-10-28

**Soundness:** 3
**Presentation:** 3
**Contribution:** 2
**Rating:** 6
**Confidence:** 4

**Summary:**

>This paper proposes DriveMamba, a novel Task-Centric and scalable State Space Model paradigm designed for efficient End-to-End Autonomous Driving. DriveMamba aims to address the limitations of conventional Transformer-based E2E-AD systems, which suffer from quadratic complexity and sequential (Perception-Prediction-Planning) design-induced cumulative errors. The core innovation is the Unified Mamba Decoder, which leverages the linear-complexity Mamba architecture to concurrently integrate dynamic Task Relation Modeling, View Correspondence Learning, and Long-term Temporal Fusion in a single stage. Crucially, DriveMamba utilizes sparse, token-level representations instead of dense BEV features and introduces a Hybrid Spatiotemporal Scan guided by the ego-vehicle's trajectory. This scanning mechanism enables efficient long-range context modeling and Ego-planning. Experimental results on the Bench2Drive and nuScenes datasets confirm that the DriveMamba-Tiny model achieves both superior performance and high efficiency, demonstrating the model's scalability and efficacy.

**Strengths:**

>S1. By replacing the quadratic-complexity Transformer with a Mamba-based decoder (SSM), the method effectively solves the major bottleneck of E2E-AD systems. This design drastically reduces memory consumption and makes the decoder easily scalable through simple layer stacking, which is a critical contribution to the exploration of scalable E2E-AD systems.

>S2. The ablation study rigorously confirms that simply stacking the decoder layers monotonically improves CIPO (Closest In-Path Objects) perception performance. This quantitatively validates that DriveMamba effectively learns perception specifically optimized for planning, rather than general scene perception.

>S3. The experiments are very dense and well-constructed. The proposed method demonstrates superior performance compared to existing baselines across the majority of in-domain scenarios.

**Weaknesses:**

>The work is well-executed, and I have only one significant concern regarding the robustness properties of the proposed architecture.

>W1. As shown in Table 10, the performance of the trajectory-guided scan appears to be highly dependent on the accuracy of the predicted trajectory. This suggests that the model might perform poorly and lack robustness in Out-of-Distribution (OOD) or extreme scenes with significant domain gaps, potentially causing planning failures. Given that robust operation in diverse and challenging deployment environments is a critical requirement for autonomous driving, the authors should have conducted more comprehensive generalization experiments (e.g., cross-dataset validation) to address this concern, which seems to be missing.

**Questions:**

>Could the authors clarify the reason for omitting the performance results of the DriveMamba-Base model in Table 2? Including this would allow for a clearer understanding of the model's scalability and the performance trend across different model sizes (Tiny, Base, Large).

---

> ### Author Response · Authors · 2025-11-21
> **Response to Reviewer RMQD**
>
> Thanks for your acknowledgement and valuable feedback, we provide the following response for your concerns:
>
> > **Q1: Robustness of the proposed architecture.**
>
> **Response:** Thanks for your kind advice and constructive comments. We exmaine the robustness of our designed framework from three main aspects:
>
> * (1) **Various Initialization of Trajectory Prior**. As default, we initialize the future waypoints of ego vehicle as "Origin", and we ablate two different initialization ways of waypoint positions, namely "Uniform" and "Random", which are iteratively refined for adaptive token sorting from local to global in diverse environments. Extensive experiments demonstrate trivial effect of different initialization methods on the performance of both ego-planning and intermediate tasks, which confirms the robustness of our codependent design for parallel processing.  For more details about experimental settings and result analysis, please refer to the **response to Reviewer KbT8 (Q1)**.
>
> * (2) **Inferior Depth Prediction**. We simulate the extreme scenes and potential failures of camera calibration and depth prediction with addition of manual Gaussian noise. Surprisingly, we observe notable robustness of DriveMamba on ego-planning performance though with inferior depth and uncertain perception. For more details about the sensitivity analysis, please refer to the **response to Reviewer KbT8 (Q4)**.
>
> * (3) **Cross-dataset Evaluation**. Besides the main experiments of DriveMamba on Bench2Drive in Tab. 3, we have also conducted the ablation of Trajectory-guided L2G scan in Task-Query B-Mamba layer for task relation modeling on Bench2Drive dataset as below:
>
> | Scan |  Driving Score| Success Rate |
> | :--- | :---: | :---: |
> | Ego-Centric L2G (Fixed) | 50.27 | 25.45 |
> | Trajectory-Centric L2G (Adaptive) | 53.54 | 27.18 |
>
> which demonstrates the effectiveness and generalizabity of our trajectory-guided scan. In the [revised version](https://openreview.net/pdf?id=MY0NHvqzi2), we include the robustness analysis section where the changes are marked in blue.
>
>
> > **Q2: Missing results of DriveMamba-Base model in Tab. 2.**
>
> **Response:** Thanks for your suggestions. Due to the limited space, we ommit the DriveMamba-Base performance in Tab. 2. And we have conducted the experiments of DriveMamba-Base on nuScenes datasets, which are shown as below:
>
> | Method | L2@1s | L2@2s | L2@3s | L2@Avg. | Coll.@1s|Coll.@2s|Coll.@3s|Coll.@Avg.| FPS |
> | :--- | :---: | :---: | :---: |  :---: | :---: |:---: |  :---: | :---: | ----- |
> | DriveMamba-Base | 0.22 | 0.40 | 0.63 | **0.41** | 0.05 | 0.06 | 0.21 | **0.11** | 6.1 |
> | DriveMamba-Base‡ | 0.17 | 0.33 | 0.54 | **0.35** | 0.02 | 0.04 | 0.17 | **0.07** | 6.1 |
>
> We can observe the consistent scalability with Tab. 3 and Tab. 6. In the [revised version](https://openreview.net/pdf?id=MY0NHvqzi2), we supplement the results of DriveMamba-Base in Tab. 2.

---

> > ### Comment · Reviewer_RMQD · 2025-11-27
> >
> > Thank you for the reply. After reviewing the author's response, I think there are no critical concerns, which leaves me with a positive outlook on the work. I will confirm the final rating after checking all the other reviews during the discussion period.

---

> > > ### Author Response · Authors · 2025-11-27
> > >
> > > Thanks for your positive feedback. We are glad that your concerns are solved and we appreciate your advice to make the manuscript better!

---

### Official Review · Reviewer_pyiN · 2025-10-31

**Soundness:** 3
**Presentation:** 3
**Contribution:** 3
**Rating:** 6
**Confidence:** 4

**Summary:**

This paper proposes DriveMamba, a Task-Centric Scalable State Space Model for efficient end-to-end autonomous driving. The core innovation lies in replacing the traditional attention-based Transformer architecture with a Unified Mamba Decoder, which jointly models perception, prediction, and planning in a single-stage pipeline with linear complexity. The authors further introduce several technical components — Hybrid Spatiotemporal Scan (HSS), task-centric tokenization, 3D sensor token localization, and long-term memory fusion — aiming to enhance scalability, efficiency, and task-level relational modeling. Experiments on nuScenes and Bench2Drive demonstrate consistent performance improvements in L2 and collision metrics, with notable inference speed.

**Strengths:**

1. Unified Mamba Decoder: Achieves linear complexity while jointly processing perception, map, and planning queries, showing clear scalability advantages on high-resolution multi-camera inputs.
2. Hybrid Spatiotemporal Scan (HSS): Cleverly alternates between spatial and ego-centric scanning to balance locality preservation and long-range temporal consistency.
3. Task-centric tokenization: Structured query design (ego/map/agent) improves modular interpretability and relational learning.
4. 3D sensor token localization: Replacing uniform ray sampling with depth-predicted projection enhances geometric accuracy and spatial reasoning.
5. Comprehensive experiments: Covers multiple benchmarks and provides results with and without ego-status, supporting generalizability and robustness claims.

**Weaknesses:**

1. Insufficient HSS details: The paper lacks explicit layer-wise configurations or stability studies when varying scan order; the contribution of each H/V-first and L2G layer is not isolated.
2. FPS reporting: Experimental FPS comparisons are unclear due to missing details on resolution, camera count, and hardware setup.
3. Depth branch robustness: No analysis of depth estimation noise, calibration error, or trade-offs between uniform-ray and learned-depth methods.
4. Trajectory prior ambiguity: The source of trajectory guidance (e.g., ego-pose history vs. future leakage) is not clarified, raising potential fairness concerns.
5. Limited interpretability of task relations: The “shared Task Query B-Mamba” strategy lacks ablation or visualization, leaving its contribution unclear.
6. Efficiency gap: The Large model exhibits a steep FPS drop, suggesting scalability bottlenecks at high model capacity.

**Questions:**

1. How is the trajectory prior in Trajectory-L2G obtained? Any risk of future information leakage?
2. How sensitive is the method to depth noise or calibration errors?
3. Could the authors clarify the HSS layer configuration and its effect on stability?

---

> ### Author Response · Authors · 2025-11-21
> **Response to Reviewer pyiN (Part 1)**
>
> Thanks for your acknowledgement and valuable feedback. Regarding your concerns, we provide the following response as below:
>
> >  **Q1: Insufficient Hybrid Spatiotemporal Scan (HSS) details about layer configuration and its effect on stability.**
>
> **Response:** Thanks for your kind advice to make the paper more comprehensive.
>
> * In Tab. 7, we first study the effect of different spatial scan types, such as Horizontal-Vertical alternation and L2G (i.e. Ego-Centric, Trajectory-Centric) **separately**. Due to the limited space, we ommit the Horizontal-First and Vertical-First ablations, which are complemented as below:
>
> | Spatial |  mAP@Det | NDS@Det |  mAP@Mapping | L2@Avg.| Coll.@Avg.|
> | :--- | :---: | :---: | :---: |  :---: | :---: |
> | Horizontal-First | 31.9 | 41.8 | 48.7 | 0.55 | 0.21 |
> | Vertical-First | 32.8 | 43.4 | 46.3 | 0.52 | 0.22 |
>
> To conclude, we find that an alternation of Horizontal-First and Vertical-First bidirectional scan, named as Horizontal-Vertical, across consecutive decoding layers (i.e. $L_{0}$: H-First, $L_{1}$: V-First, $L_{2}$: H-First, ..., etc) can preserve better spatial locality for **view correspondence learning** (Line 231), leading to accurate perception. Besides, Trajectory-Centric L2G spatial scan exhibits superior planning performance from ego perspective during the **task relation modeling**. While the spatial-first temporal scan excels at **long-term temporal fusion**. Intuitively, we combine these three scan strategies into a unified decoder for different purposes of B-Mamba layers, termed as hybrid spatiotemporal scan. In the [revised version](https://openreview.net/pdf?id=MY0NHvqzi2), we add more details about this part and supplement more ablations as well as explanations for comprehension.
>
> * As for the effect on stability, we find that the alternation of H/V-First spatial scan contributes to the training convergence and stability of DriveMamba for structural representation learning from sensor tokens, thus providing high-quality task queries for relation modeling and temporal fusion. Besides, Trajectory-Centric L2G scan can further facilitate ego-planning with adapative interaction order. Moreover, each ablation about hybrid scan combination has been carried out with three repeated experiments independently, demonstrating the consistent conclusion. In the [revised version](https://openreview.net/pdf?id=MY0NHvqzi2), we add more discussion about the effect on stability (changes are marked in blue).
>
>
>
> > **Q2: How is the trajectory prior obtained and concern about the future information leakage.**
>
> **Response:** Thanks for your suggestions. Specifically, we initialize the ego query with a series of waypoint tokens during  the task-centric tokenization process, where each token encodes a discrete waypoint clue (Line 304-306). Thanks to the parallel processing and iterative optimization designs of DriveMamba, the future waypoints of ego vehicle are iteratively refined from coarse to fine across decoding layers, which provide dynamic trajectory prior and enable adaptive token sorting without any future information leakage. And we study the training stability of DriveMamba with different waypoint initialization ways of ego vehicle, demonstrating the robustness of DriveMamba. For more details, please refer to the **response to Reviewer KbT8 (Q1)**. In the [revised version](https://openreview.net/pdf?id=MY0NHvqzi2), we include these details and analysis which are highlighted in blue.
>
>
> > **Q3: Trade-offs with uniform-ray methods and robustness of depth prediction branch.**
>
> **Response:** Thanks for your valuable suggestions.
>
> * Unlike transformer-based methods generate 3D position-aware features through **uniform-ray** position embeddings, which neglects the actual depth distribution and can not be directly employed for token sorting in 3D space. Differently, we resort to the **3D point** positional encoding which is indispensable and could provide more precise yet efficient (ray vs. single point) position information by encoding the location of a single point with an estimated depth.
> * As for the sensitivity of DriveMamba to depth prediction errors, we have carried out the comprehensive robustness analysis of depth quality, showcasing the capacity of DriveMamba in handling inferior depth prediction results. For more details, please refer to the **response to Reviewer KbT8 (Q4)**.

---

> ### Author Response · Authors · 2025-11-21
> **Response to Reviewer pyiN (Part 2)**
>
> >  **Q4: Missing details on resolution, camera count and hardware setup when reporting FPS.**
>
> **Response:** Thanks for your efforts in improving our paper quality. For the resolution of different variants of DriveMamba, we report them as shown in Tab. 1. And all compared methods adopt 6 cameras as multi-view input. For the detailed comparison with other methods, please refer to the following table:
>
> | Method   | Resolution | FPS | Hardware |
> | :---- | :-----: | :-----: |  :-----: |
> | UniAD | 1408x512 | 1.8  | Tesla A100|
> | VAD-Tiny |  640x360| 16.8   | RTX 3090|
> | VAD-Base |  1280x720| 4.5   |RTX 3090|
> | DriveTransformer-S | 1056x384   | 10.7 | A800|
> | DriveTransformer-B   | 1056x384   | 7.2 | A800|
> | DriveMamba-T   | 704x256   | 17.9 | RTX 3090|
> | DriveMamba-S   | 1056x384  | 11.1 | RTX 3090|
>
> In the final version, we will add the detailed configurations in the table.
>
>
> >  **Q5: Limited interpretability of task relations.**
>
> **Response:** Thanks for valuable feedback. We would like to clarify that there are three types of B-Mamba layers inside each unified decoder block for different purposes (i.e. Task-Sensor semantic extraction, Task-Memory temporal fusion and **Task-Task interaction modeling**). And the Task-Query B-Mamba layer performs task-relation modeling among different types of task queries (i.e. agent, map and ego) following the Trajectory-Centric L2G scan order **as described in the reponse to Q1**. It is important and necessary to adopt such operation for task synergy learning, such as interactive planning and planning-oriented perception as validated in Tab. 5. To further study the effect of this layer, we add the ablation study shown as below:
>
> | Method |  Driving Score| Success Rate |
> | :---- | :-----: | :-----: |
> | Full Attention | 53.54 | 27.18 |
> | w/o Task Relation | 45.45 | 25.00 |
>
> which demonstrate that the task relation modeling layer is helpful to improve the driving score since the ego query could utilize the detected objects and map elements to conduct interactive planning. In the final version, we will highlight the design motivation and contribution of each design choice with sufficient ablation.
>
> > **Q6: Efficieny gap with large model.**
>
> **Response:** Thanks for your question. In fact, the steep FPS drop of DriveMamba-Large mainly results from the deeper visual encoder (ViT-L/16). However, the efficiency scaling trend of unified decoder of DriveMamba is relatively gentle, allowing more stacked layers for further model scaling. We will clarify the efficiency gap of large model in the final version.

---

### Official Review · Reviewer_KbT8 · 2025-11-04

**Soundness:** 3
**Presentation:** 3
**Contribution:** 3
**Rating:** 6
**Confidence:** 4

**Summary:**

This paper identifies key challenges in current End-to-End Autonomous Driving (E2E-AD) systems. It argues that dominant methods, which often use sequential Transformer decoders (e.g., perception-prediction-planning), suffer from information loss, cumulative errors, and inflexible task modeling. Furthermore, the reliance on dense BEV features and the quadratic complexity of Transformer-based attention mechanisms create bottlenecks in efficiency and scalability. To address these issues, the paper proposes DriveMamba, a "Task-Centric Scalable paradigm" for E2E-AD. The central idea is the replacement of Transformer decoders with a Unified Mamba decoder. This decoder is based on State Space Models (SSMs), which have linear complexity, to improve efficiency and scalability.

DriveMamba is a single-stage, parallel framework. It tokenizes multi-view images and task-specific queries (Agent, Map, Ego) into sparse representations. These tokens, along with positional embeddings derived in part from a predicted depth map, are fed into the unified decoder. This decoder is designed to simultaneously learn view correspondence, dynamic task relations, and long-term temporal fusion.
A key component is the hybrid spatiotemporal scan method, which is required to apply the 1D Mamba model to the 3D driving scene. The paper introduces a "bidirectional trajectory-guided 'local-to-global' scan". This method dynamically sorts tokens based on their proximity to an intermediate predicted ego-trajectory, aiming to preserve spatial locality from an ego-centric perspective.

Experiments are conducted on the nuScenes and Bench2Drive datasets for both open-loop and closed-loop evaluation. The results show that DriveMamba models achieve lower L2 displacement error and collision rates compared to previous methods, while also demonstrating significant improvements in inference speed (FPS) and reductions in GPU memory consumption. The paper includes ablation studies on the decoder's modular components , scan methods , and scalability.

**Strengths:**

- The paper clearly articulates two significant problems in E2E-AD: 1) The limitations of sequential, manually-ordered pipelines, such as information loss and error accumulation. 2) The efficiency and scalability constraints imposed by the quadratic complexity of attention in Transformer models.
-  The idea to replace the Transformer decoder with a Mamba-based (SSM) decoder  directly addresses the efficiency and scalability problem. The linear complexity of SSMs is a clear advantage for processing long spatiotemporal sequences. Figure 5 provides a clear comparison of this, showing a 3.2x speed increase and 68.8% less memory use at higher resolutions.
- The paper proposes a "Trajectory-Centric Local2Global" (TC-L2G) scan method. The idea of dynamically sorting tokens based on their relevance to the predicted future ego-path is an explicit attempt to inject an ego-centric bias, which is relevant for the planning task.
- The approach moves away from dense BEV feature maps and instead uses sparse tokenized representations for both sensor inputs and task outputs. These are processed in parallel by a unified decoder, which is designed to enable dynamic modeling of task relationships.
- The model is evaluated on both open-loop (nuScenes) and closed-loop (Bench2Drive)  benchmarks. This dual evaluation provides a more complete picture of the model's planning capabilities, as open-loop metrics do not always correlate with real-world driving performance.

**Weaknesses:**

I have the following concerns from this work

- The "Trajectory-Centric Local2Global" scan (L232) creates a potential circular dependency. The scan order, which is an input to the decoder layers, is determined by an importance weight $w_i$ calculated from an intermediate predicted ego-trajectory $\psi^{\prime}$. This means the decoder's output (the trajectory) is required to define its input (the scan order). The paper does not specify how this intermediate trajectory is generated or analyze the stability of this co-dependent design. Table 10  shows that using a Ground-Truth trajectory improves results, but this does not resolve the question of how the model functions in practice.

- The scalability study in Table 6 presents conflicting data. As the decoder is scaled from 3 to 12 layers, closed-loop planning performance improves (51.1 to 66.5). However, open-loop perception performance (Detection mAP, NDS, and Mapping mAP) decreases (e.g., Detection mAP drops from 34.8 to 33.1, Mapping mAP drops from 50.3 to 46.6). This suggests a performance trade-off between tasks, not uniform scalability. The paper's explanation that the model learns "planning-oriented perception" is an interpretation that requires more evidence, as it implies general perception is being sacrificed.
- To support the "planning-oriented perception" claim, the paper provides Table 5 showing that perception of "Closest In-Path Objects (CIPO)" improves with more decoder layers. However, this table only shows CIPO metrics. It does not show the general perception metrics for the same models. To make a convincing argument, the paper should present both CIPO and general perception metrics side-by-side to demonstrate that while general perception degrades (as suggested by Table 6), CIPO-specific perception improves. The current presentation disconnects these two key results.
- The 3D position of sensor tokens, $P_{sensor}$, is a critical component for token sorting and view correspondence. This position is entirely dependent on a predicted depth value $d_{i,k}$, which comes from an auxiliary depth prediction branch. This introduces a significant potential failure point. If the predicted depth is inaccurate, the 3D positions of sensor tokens will be incorrect, leading to flawed token sorting and feature extraction. An ablation study on the accuracy of this depth branch or the sensitivity of the overall model to depth prediction errors is needed.

**Questions:**

Please see weakness above

---

> ### Author Response · Authors · 2025-11-21
> **Response to Reviewer KbT8 (Part 1)**
>
> Thanks for your acknowledgement and valuable feedback, we provide the following response for your concerns:
>
> >  **Q1: How the intermediate trajectory is generated and missing stability analyis of this codependent design.**
>
> **Response:** Thanks for the kind advice and detailed question.
>
> * First, as stated in the token initialization section (Line 178), we would like to clarify that each pre-defined task query is equipped with both semantic embeddings and positional embeddings. And we initialize the BEV location of ego query as origin (0, 0) (Line 188). Specifically, DriveMamba employs a series of tokens (e.g. $T_{e}$ = 6) to represent future waypoints of ego vehicle, where each token encodes a discrete waypoint position (Line 304-306). Then the future waypoints are iteratively refined with predicted offsets across consecutive decoding layers through planning head (Line 307-309), facilitating the adaptive token sorting from ego-perspective. In the [revised version](https://openreview.net/pdf?id=MY0NHvqzi2), we add more details about the intermediate trajectory generation for clarity.
>
> * Second, as for the stability of codependent design, we try to ablate different initialization ways of future waypoints used for agent/map query sorting in the first decoding layer, and the results are shown as below:
>
> | Initialization | mAP@Det | NDS@Det |  mAP@Mapping | L2@Avg.| Coll.@Avg.|
> | :--- | :---: | :---: | :---: |  :---: | :---: |
> | Origin | 34.8 | 45.4 | 50.3 | 0.44 | 0.15 |
> | Uniform | 34.7 | 45.3 | 50.3 | 0.46 | 0.16 |
> | Random | 34.6 | 45.1 | 50.2 | 0.45 | 0.14 |
>
> where  "Origin", "Uniform" and "Random" indicate zero offset, fixed offset ($\triangle_{x}$=0, $\triangle_{y}$=1) and random offset ( $\mu$=0,  $\sigma$=1) initialization of future waypoints of ego vehicle, respectively. And we can find that though with different initialized waypoints, both perception and planning performance exhibit trivial changes, showcasing the training stability of DriveMamba and robustness of our proposed egocentric design for task synergy learning. In the [revised version](https://openreview.net/pdf?id=MY0NHvqzi2), we supplement the stability analysis of such codependent design.
>
> > **Q2: Missing general perception results in Tab. 5.**
>
> **Response:** Thanks for your detailed suggestions.  Actually, the general perception performance of DriveMamba with ResNet-50 backbone and different number of decoding layers are shown in Tab. 6, which **result from the same models in Tab. 5 accordingly**. Due to the limited space, we split them into two tables for separate discussions. In the final version, we will add more explanations and consider merging these two tables to avoid confusion.

---

> ### Author Response · Authors · 2025-11-21
> **Response to Reviewer KbT8 (Part 2)**
>
> > **Q3: More evidence for "planning-oriented perception" during the scalability study in Tab. 6.**
>
> **Response:** Similar to **Q2**, we would like to clarify that the "planning-oriented perception" results of DriveMamba with ResNet-50 and different number of decoding layers are shown in Tab. 5 respectively, which confirm the consistent scalability of our unified planning design for intermediate tasks. And we believe that such ego-centric task synergy learning is more intuitive, interpretable and efficient,  meanwhile conforms to the natural attention distribution of human drivers. Moreover, **we appeal the research community to adopt our proposed CIPO metric to measure the perception performance for end-to-end driving applications if possible**.
>
>
>
> > **Q4: Sensitivity of overall model to depth prediction errors.**
>
> **Response:** Thanks for your valuable suggestions. As shown in the following table, we further evaluate the sensitivity of overall model to various qualities of depth predictor, which is indispensable for sensor token sorting in the view correspondence learning layer during unified decoding. In short, when utilizing the Ground-Truth depth map for sensor token positioning, our DriveMamba can obtain noticeable improvement of perception performance (**+4.9 NDS**) as expected. However, we observe relatively smaller improvement of planning performance (**-0.02% Collision Rate**), demonstrating the robustness of our DriveMamba in handling inaccurate or uncertain detections. Besides, when adding the Gaussian noise to either camera extrinsic parameters (incorrect calibration) or predicted depth map directly, the perception performance drops more significantly than planning performance respectively. It might be because DriveMamba avoids the sequential modeling and construction of BEV features as previous methods, which are sensitive to perception inputs. On the contrary, DriveMamba directly interacts with raw sensor features and thus be able to ignore those failure or noisy inputs and demonstrates better robustness on ego-planning. In the [revised version](https://openreview.net/pdf?id=MY0NHvqzi2), we include this ablation and add more robustness analysis.
>
>
> | Depth |  mAP@Det | NDS@Det |  mAP@Mapping | L2@Avg.| Coll.@Avg.|
> | :--- | :---: | :---: | :---: |  :---: | :---: |
> | Incorrect Calibration | 31.7 | 43.1 | 47.5 | 0.46 | 0.16 |
> | Noisy Prediction | 32.4 | 43.9 | 48.1 | 0.45 | 0.15 |
> | Normal Prediction     | 34.8 | 45.4 | 50.3 | 0.44 | 0.15 |
> | GT | 42.3 | 50.5 | 54.8 | 0.42 | 0.13 |

---

### Meta-Review · Area_Chair_fD5k · 2025-12-25

**Summary:**

The paper proposes DriveMamba, a Task-Centric Scalable framework for End-to-End Autonomous Driving. It replaces standard Transformer decoders with a Unified Mamba Decoder based on State Space Models to address the quadratic complexity bottlenecks of attention mechanisms in long-sequence processing. To adapt the 1D Mamba architecture to 3D driving scenarios, the authors introduce a Hybrid Spatiotemporal Scan and a Trajectory-Guided Local-to-Global scanning strategy. The method utilizes sparse token representations for sensors and tasks, achieving competitive planning performance on nuScenes and Bench2Drive benchmarks with significantly improved inference speed and reduced memory consumption.

Overall, the reviewers hold a positive attitude towards this paper (all gave a score of 6). I believe that introducing linear models like Mamba to improve efficiency is significant, particularly in the domain of autonomous driving where efficient inference is critical. However, I also share Reviewer qoTq's concerns regarding the novelty. Ultimately, I have decided to accept this paper as a Poster.

**Reviewer Concerns:**

All concerns raised by reviewers were effectively addressed during the discussion phase:

- **Robustness and Circular Dependency**: Reviewers KbT8, pyiN, and RMQD raised concerns about the "Trajectory-Guided" scan creating a circular dependency (since the trajectory is both input and output) and the system's sensitivity to depth prediction errors. The authors provided convincing ablation studies showing that the model is robust to initialization strategies (Origin/Uniform/Random) due to iterative refinement, proving the design is stable. Adding Gaussian noise to depth maps affects perception metrics but leaves planning performance relatively stable, demonstrating the robustness of the end-to-end planning capability.

-  **Novelty**: Reviewer qoTq noted the framework's similarity to DriveTransformer and questioned if the contribution was merely replacing "Attention with Mamba." The authors and other reviewers acknowledged that the specific scanning designs (L2G) required to make SSMs work for 3D geometric tasks constitute a non-trivial and valuable engineering contribution.

- **Perception vs. Planning Trade-off**: Reviewer KbT8 noted that general perception metrics dropped while planning improved. The authors successfully argued that "Planning-oriented Perception" (evidenced by improved CIPO metrics) is the correct objective for E2E-AD, a justification the reviewers accepted.

- **Missing Details**: Reviewer pyiN's concerns regarding missing hardware details for FPS comparisons were resolved by supplementary tables provided during the rebuttal.

**Reviewer Scores:**

The authors' rebuttal has addressed most of the reviewers' concerns. Although some reviewers still hold reservations, they have explicitly expressed a positive attitude towards the paper.
- Reviewer KbT8: 6 => 6.
- Reviewer pyiN: 6 => 6.
- Reviewer RMQD: 6 => 6.
- Reviewer qoTq: 6 => 6.

---

### Decision · Program_Chairs · 2026-01-26

Accept (Poster)